# Training Scale-Invariant Neural Networks on the Sphere Can Happen in Three Regimes

**Maxim Kodryan**[1]*, **Ekaterina Lobacheva**[1]*, **Maksim Nakhodnov**[2]*, **Dmitry Vetrov**[1,2]

[1]HSE University     [2]AIRI

Moscow, Russia

mkodryan@hse.ru, elobacheva@hse.ru, nakhodnov@2a2i.org, dvetrov@hse.ru

## Abstract

A fundamental property of deep learning normalization techniques, such as batch normalization, is making the pre-normalization parameters scale invariant. The intrinsic domain of such parameters is the unit sphere, and therefore their gradient optimization dynamics can be represented via spherical optimization with varying effective learning rate (ELR), which was studied previously. However, the varying ELR may obscure certain characteristics of the intrinsic loss landscape structure. In this work, we investigate the properties of training scale-invariant neural networks directly on the sphere using a fixed ELR. We discover three regimes of such training depending on the ELR value: convergence, chaotic equilibrium, and divergence. We study these regimes in detail both on a theoretical examination of a toy example and on a thorough empirical analysis of real scale-invariant deep learning models. Each regime has unique features and reflects specific properties of the intrinsic loss landscape, some of which have strong parallels with previous research on both regular and scale-invariant neural networks training. Finally, we demonstrate how the discovered regimes are reflected in conventional training of normalized networks and how they can be leveraged to achieve better optima.

## 1 Introduction

Most modern neural network architectures contain some type of normalization layers, such as Batch Normalization (BN) [10] or Layer Normalization [2]. Normalization makes networks partially *scale-invariant* (SI), i.e., multiplication of their parameters preceding the normalization layers by a positive scalar does not change the model's output. In general, the training dynamics of model parameters can be viewed from two perspectives: the *direction dynamics*, i.e., the dynamics of the projection of the parameters onto the unit sphere, and the *norm dynamics*. For SI neural networks, the former seems to be more important since the direction alone determines the output of the scale-invariant function. However, the latter influences the optimization by changing the *effective learning rate* (ELR), i.e., the learning rate (LR) a scale-invariant model would have if it were optimized on the unit sphere.

Many works have studied the effect of scale invariance on training dynamics through the prism of norm or, equivalently, ELR dynamics [33, 9, 1, 38, 24, 26, 29, 35, 27]. They discovered that during standard training with weight decay the ELR can change in a non-trivial way and lead to different optimization dynamics: periodic behavior [27], destabilization [22, 24], or stabilization on some sphere [26, 35]. However, considering only the ELR dynamics is not enough for a comprehensive study of the specifics of training normalized neural networks. In particular, the impact of the varying ELR on training may obscure certain important properties of SI neural networks' intrinsic domain, i.e., the unit sphere, thus narrowing our intuition about the loss landscape structure. Therefore, in this work, we are focused on the direction dynamics. To eliminate the influence of the changing norm on

---

*First three authors contributed equally.

36th Conference on Neural Information Processing Systems (NeurIPS 2022).

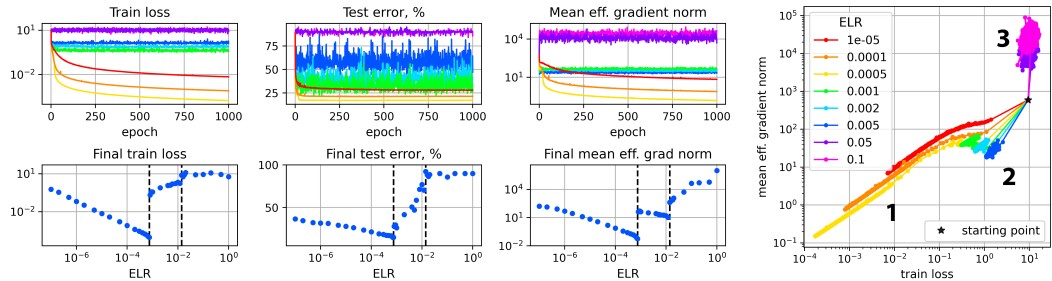

Figure 1: Three regimes of SI neural network training on the sphere: (1) convergence for the lowest ELRs, (2) chaotic equilibrium for medium ELRs, and (3) divergence for the highest ELRs. ConvNet on CIFAR-10. Dashed lines on scatter plots denote borders between the regimes.

training, we fix the total norm of all scale-invariant parameters of the normalized neural network and train it with a constant effective learning rate on the sphere.

We discover three regimes of such an optimization procedure depending on the ELR value (see Figure 1). The first regime (small ELRs) can be considered as a typical convergence to a minimum with monotonically decaying training loss. The second one (medium ELRs) demonstrates a sustained oscillating behavior of the loss around a certain value separated from both the global minimum and the random guess behavior. We call this regime a *chaotic equilibrium*, reminiscent of the equilibrium state reported in several previous works [26, 35]. The last regime (high ELRs) represents destabilized, diverged training associated with an excessively large optimization step size.

We analyze the underlying mechanisms behind each of these regimes on a toy example and conduct an extensive empirical study with neural networks trained on the sphere using projected stochastic gradient descent (SGD). We point out the main features of each regime and study how they relate to each other and what peculiarities of the optimization dynamics and intrinsic structure of the loss landscape they reveal. Our findings both reconfirmed some of the previous beliefs, like that training with larger (E)LR generally ends up in better optima, or that high-sharpness zones are present in the optimization trajectory, and introduced novel insights, like that different (E)LRs of the first regime even with the same initialization and batch ordering actually lead to distinguishable minima with different sharpness/generalization profiles depending on the specific (E)LR value, or that the high-sharpness zones are responsible for the boundary between the first and the second regimes and have close ties with the Double Descent phenomenon. Finally, we demonstrate how our results are reflected in conventional neural network training and suggest several practical implications that help achieve optimal solutions in terms of sharpness and generalization.

Our code is available at `https://github.com/tipt0p/three_regimes_on_the_sphere`.

## 2 Background and related work

As mentioned in the introduction, for every function $F(\boldsymbol{\theta})$ optimized w.r.t. its parameters $\boldsymbol{\theta}$ we can separate the dynamics of the parameters direction $\boldsymbol{\theta}/\|\boldsymbol{\theta}\|$ from the dynamics of their norm $\|\boldsymbol{\theta}\|$. If the function $F(\boldsymbol{\theta})$ is scale-invariant, i.e., $F(c\boldsymbol{\theta}) = F(\boldsymbol{\theta})$, $\forall c > 0$, the former is the only factor that determines the value of the function, since $F(\boldsymbol{\theta}/\|\boldsymbol{\theta}\|) = F(\boldsymbol{\theta})$. However, even though the SI function's value does not depend on the parameters norm, the latter greatly influences its gradients: $\nabla F(c\boldsymbol{\theta}) = \nabla F(\boldsymbol{\theta})/c$, $\forall c > 0$ and thus the optimization speed. This argument can be equivalently reformulated in terms of how the norm dynamics control the direction dynamics' speed via the notion of effective learning rate. For the standard (S)GD optimization, ELR is defined as the learning rate divided by the squared parameters norm, and it represents how the learning rate should be adapted to mimic the same direction dynamics if the function were optimized directly on the unit sphere [29].

Most of the previous work has dealt with the implications of using normalization in neural networks, i.e., making them (partially) scale-invariant, from the perspective of the ELR dynamics. For example, Arora et al. [1] study the convergence properties of batch normalized neural networks and explain why they allow much higher learning rates compared to their non-BN counterparts. Many works have also focused on training scale-invariant neural networks with weight decay [33, 38, 24, 26, 35, 27].

They all discovered the non-trivial dynamics of ELR induced by the interplay between normalization and weight decay but came to different conclusions. Some studies have reported training instability caused by this interplay [22, 24], while others have argued that, on the contrary, the equilibrium state is eventually reached, i.e., ELR becomes stable after some time [33, 4, 26, 20, 35]. Recently, Lobacheva et al. [27] demonstrated that training with weight decay and constant LR may experience regular periodic behavior for a practical range of LRs.

However, while all of these works mainly study the direction dynamics with varying rate, they lack exploring the properties of optimization of SI models with a *fixed* effective learning rate. Decoupling the direction dynamics from the changing norm effects could reveal particular features of the intrinsic domain of scale-invariant neural networks that were potentially obscured by the dynamical ELR. To cope with it, we optimize our models directly on the sphere using projected SGD with a constant learning rate, which is equivalent to fixing the ELR. Our method reveals some interesting results about SI neural networks and allows to draw parallels with previous scattered deep learning work. We note that similar experiments were conducted by, e.g., Arora et al. [1] and Lobacheva et al. [27] to highlight certain features of standard training when ELR is dynamical. Also, Cho and Lee [5] optimized scale-invariant parameters in normalized neural networks directly on their intrinsic domain using Riemannian gradient descent, but they proposed an optimization method rather than investigating the properties of such training.

## 3  Theoretical analysis

To explain the observed behavior, we proceed with the theoretical analysis of scale-invariant functions optimization with a fixed effective learning rate. At first, we derive several important general properties of such a process that will help to clarify some of its empirical aspects. Then we consider a concrete example of a function with multiple scale-invariant parameter groups, similar to normalized neural networks, and study its convergence depending on the ELR value to shed more light on the prime causes of the three regimes. We provide all formal derivations and proofs in Appendix A.

### 3.1  General properties

We begin with an analysis of common features of all SI functions optimization with a fixed learning rate on the sphere. For that, consider a function $F(\boldsymbol{\theta})$ of a vector parameter $\boldsymbol{\theta} \in \mathbb{R}^P$ that can be divided into $n$ groups: $\boldsymbol{\theta} = (\boldsymbol{\theta}_1, \ldots, \boldsymbol{\theta}_n)$, where each $\boldsymbol{\theta}_i \in \mathbb{R}^{p_i}$ and $\sum_{i=1}^n p_i = P$. We say that each of these groups is scale-invariant, i.e., we can multiply it by a positive scalar value, with the others fixed, and the output of the function does not change. This is typical for neural networks with multiple normalized layers: each of these layers (and even each column/filter in a layer) is scale-invariant. Naturally, if a function is scale-invariant w.r.t. several parameter groups, it is also scale-invariant w.r.t. their union, therefore the whole vector $\boldsymbol{\theta}$ is scale-invariant too.

Since scale-invariant functions are effectively defined on the sphere, we try to minimize $F(\boldsymbol{\theta})$ on the sphere of radius $\rho$:

$$\min_{\boldsymbol{\theta} \in \mathcal{S}^{P-1}(\rho)} F(\boldsymbol{\theta}), \tag{1}$$

where $\mathcal{S}^{P-1}(\rho) = \{\theta \in \mathbb{R}^P : \|\theta\| = \rho\}$. We do that using the projected gradient descent method with a fixed learning rate $\eta$:

$$\begin{cases} \hat{\boldsymbol{\theta}}^{(t)} \leftarrow \boldsymbol{\theta}^{(t)} - \eta \nabla F(\boldsymbol{\theta}^{(t)}), \\ \boldsymbol{\theta}^{(t+1)} \leftarrow \hat{\boldsymbol{\theta}}^{(t)} \cdot \frac{\rho}{\|\hat{\boldsymbol{\theta}}^{(t)}\|}. \end{cases} \tag{2}$$

Of course, since the function is scale-invariant w.r.t. each individual group $\boldsymbol{\theta}_i$, we could optimize it on the product of $n$ independent spheres $\mathcal{S}^{p_i-1}$ instead of the single mutual sphere, as we do in eq. (1). However, that would either require introducing too many hyperparameters in the model ($n$ different learning rates for $n$ spheres) or overconstrained the problem (one learning rate for all $n$ spheres). We also argue that constraining the total parameters norm is much more natural and closer to the standard neural networks training with weight decay (see discussion in Appendix A.1).

We now define the notions of *effective gradient* and *effective learning rate* for each SI group $\boldsymbol{\theta}_i$. Effective gradient is the gradient w.r.t. $\boldsymbol{\theta}_i$ measured at the same point but with the group $\boldsymbol{\theta}_i$ projected on its unit sphere, i.e., at point $\tilde{\boldsymbol{\theta}} = (\boldsymbol{\theta}_1, \ldots, \boldsymbol{\theta}_i/\rho_i, \ldots, \boldsymbol{\theta}_n)$, where $\rho_i \equiv \|\boldsymbol{\theta}_i\|$. Due to scale

invariance w.r.t. $\boldsymbol{\theta}_i$, the function's value is equal at both points $\boldsymbol{\theta}$ and $\tilde{\boldsymbol{\theta}}$ but, as discussed earlier, the norm of the effective gradient, which we denote as $\tilde{g}_i$, is $\rho_i$ times the norm of the regular gradient w.r.t. $\boldsymbol{\theta}_i$: $\tilde{g}_i = g_i\rho_i$, where $g_i \equiv \|\nabla_{\boldsymbol{\theta}_i} F(\boldsymbol{\theta})\|$. Effective learning rate $\tilde{\eta}_i$ is the learning rate required to make a gradient step w.r.t. $\boldsymbol{\theta}_i$ equivalent to the original but started from the point $\tilde{\boldsymbol{\theta}}$. It is known from the previous literature on scale-invariant functions [29, 35, 27] that ELR for the group $\boldsymbol{\theta}_i$ equals $\tilde{\eta}_i = \eta/\rho_i^2$. We also define the total effective gradient norm $\tilde{g} = g\rho$, where $g \equiv \|\nabla_{\boldsymbol{\theta}} F(\boldsymbol{\theta})\|$, and the total ELR $\tilde{\eta} = \eta/\rho^2$, which is *fixed* in the considered setup.

From the constraint on the total norm $\sum_{i=1}^n \rho_i^2 = \rho^2$ we obtain the following fundamental equation relating the ELRs of individual SI groups to the total ELR value:[2]

$$\sum_{i=1}^n \frac{1}{\tilde{\eta}_i} = \frac{1}{\tilde{\eta}}. \tag{3}$$

Another important notion is the *effective step size* (ESS), which is the product of the effective gradient length by the effective learning rate: $\tilde{\eta}_i\tilde{g}_i$. This value defines how far we effectively move the scale-invariant parameters $\boldsymbol{\theta}_i$ after one gradient step. It can be shown that the total squared ESS can be expressed as a convex combination of squared ESS values of individual SI groups:

$$(\tilde{\eta}\tilde{g})^2 = \sum_{i=1}^n \omega_i(\tilde{\eta}_i\tilde{g}_i)^2, \ \sum_{i=1}^n \omega_i = 1, \ \omega_i \propto \frac{1}{\tilde{\eta}_i}. \tag{4}$$

The process (2) and eq. (3) yield the following update rule for each group's individual ELR:

$$\tilde{\eta}_i^{(t+1)} \leftarrow \tilde{\eta}_i^{(t)} \frac{1 + (\tilde{\eta}\tilde{g}^{(t)})^2}{1 + (\tilde{\eta}_i^{(t)}\tilde{g}_i^{(t)})^2}, \tag{5}$$

from which it follows that *for a given SI group the higher (lower) the ESS at the current iteration, the lower (higher) the ELR becomes after it*. Since, by definition, ESS and ELR values are highly correlated, we conclude that at each iteration the largest ELRs tend to become smaller and vice versa. This "negative feedback" principle becomes very important in distinguishing between the three regimes of optimization.

## 3.2  Explaining the regimes

To provide a clear and easy-to-follow explanation of the differences between the three regimes of training on the sphere, we construct a simple yet illustrative example of a function with several groups of SI parameters and analyze its optimization properties depending on the total ELR value.

The simplest working example of a scale-invariant function is the following one:

$$f(x,y) = \frac{x^2}{x^2 + y^2}, \ x, y \in \mathbb{R} \setminus \{0\}. \tag{6}$$

This function has also been used in previous works to visually demonstrate various properties of general scale-invariant functions [26, 27]. Based on it, we develop an elucidative example of a function that contains more than one group of scale-invariant parameters — just as real multilayer normalized neural networks do. For that, we take a conical combination of $n > 1$ functions (6) each depending on its own pair of variables:

$$F(\boldsymbol{x}, \boldsymbol{y}) = \sum_{i=1}^n \alpha_i f(x_i, y_i) = \sum_{i=1}^n \alpha_i \frac{x_i^2}{x_i^2 + y_i^2}, \tag{7}$$

where $\boldsymbol{x} = (x_1, \ldots, x_n)$, $\boldsymbol{y} = (y_1, \ldots, y_n)$, and each $\alpha_i > 0$ is a fixed positive coefficient. In this function, there are $n$ groups of SI parameters: $(x_i, y_i), i = 1, \ldots, n$. In accordance with the setup described previously, we intend to optimize this function on the unit sphere using the projected gradient descent method with a fixed (effective) learning rate, i.e., we plug $\boldsymbol{\theta} = (\boldsymbol{x}, \boldsymbol{y}) = ((x_1, y_1), \ldots, (x_n, y_n))$, $P = 2n$, $\rho = 1$ into equations (1) and (2).

Now we would like to study the behavior of optimization of $F(\boldsymbol{x}, \boldsymbol{y})$ on the unit sphere depending on the learning rate value $\eta$, which equals to the total ELR $\tilde{\eta}$ due to $\rho = 1$. But first we need to dwell on the convergence properties of each of its subfunctions $\alpha_i f(x_i, y_i)$. The next proposition, which we rigorously formulate and prove in Appendix A.3, answers this question.

---

[2]See the derivation of this and the following equations of this section in Appendix A.2.

**Proposition 1** *For the SI function $f_\alpha(x, y) = \alpha \frac{x^2}{x^2+y^2}$ minimized with ELR $\tilde\eta$:*

1. *if $\tilde\eta < \frac{1}{\alpha}$, it linearly converges to zero, i.e., its global minimum;*

2. *if $\tilde\eta > \frac{1}{\alpha}$, it stabilizes at value $\frac{1}{2}\left(\alpha - \frac{1}{\tilde\eta}\right)$.*

Knowing that each of the subfunctions of $F(\boldsymbol{x}, \boldsymbol{y})$ can converge to the minimum only when its ELR stays below $1/\alpha_i$ and considering the fundamental constraint on ELRs (3), we state that *the first regime (convergence)* is only possible when the total ELR is below a certain threshold:

$$\tilde\eta < \frac{1}{\sum_{i=1}^{n} \alpha_i}. \tag{8}$$

What if the total ELR exceeds that threshold? The first regime cannot be observed: convergence becomes impossible, since at least one of the individual ELRs exceeds its convergence threshold value, according to eq. (3). By Proposition 1 the value of the corresponding subfunction is separated from zero by some quantity depending on its ELR. This means that the total function value, and therefore the total effective gradient, is also nonvanishing (we give a more formal argument in Appendix A.4). This brings us to the "undamped" dynamics of ELRs (5), where the effective step sizes do not tend to zero. In that case, the dynamics stabilize only when all the ESS values become equal, which is possible if and only if each individual ELR $\tilde\eta_i$ takes its corresponding *equilibrium* value:

$$\tilde\eta_i^* \equiv \frac{\tilde\eta \sum_{j=1}^{n} \alpha_j}{\alpha_i}. \tag{9}$$

Due to the negative feedback principle of eq. (5), the system will naturally tend towards this state when ESS values are non-negligible. Note that, in contrast, in the first regime, ESS values decay too rapidly due to convergence, and the model may fail to reach equilibrium. In sum, when the total ELR exceeds the convergence threshold and each individual ELR (in average) equals its equilibrium value corresponding to the equalized ESS, the optimization process enters *the second regime, i.e., the chaotic equilibrium*. Finally, if the total ELR is set too high, the chaos will dominate equilibration and the optimization dynamics will resemble a random walk. In that case, we observe *the third regime, or divergence*.

In Figure 2, we plot the evolution of the function (7) during optimization on the unit sphere with three different effective learning rates. We set $n = 3$, $\alpha_i = 2^i$, $i = 0, 1, 2$, and start from a point randomly selected on the 6-dimensional unit sphere. In this case, the threshold condition (8) becomes $\tilde\eta < 1/7 \approx 0.143$, and thus we select three ELRs corresponding to the three optimization regimes, respectively: $0.1$, $0.2$, and $0.5$. The lines on the plot directly demonstrate the anticipated behavior: the smallest ELR leads to a fast decay of the function's value, the medium one makes the function stabilize around a certain level, and the largest ELR depicts the most chaotic behavior. In Appendix A.5, we provide plots of the individual ELRs dynamics that show that in the chaotic equilibrium $\tilde\eta_i$ in average by iterations closely match the predicted values (9), while in the first regime they are not reached due to quick convergence, and in the third the optimization is too unstable to distinguish any equilibration.

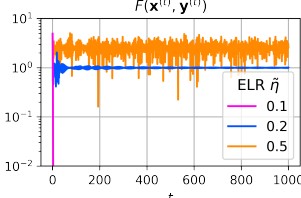

Figure 2: Function values evolution during optimization of the function (7) using three different ELRs corresponding to the three optimization regimes, respectively.

## 4 Experimental setup

After discussing the theoretical substantiation of the different regimes of learning on the sphere in the case of a simple SI function, let us move on to an empirical analysis of the learning dynamics of neural networks equipped with Batch Normalization. We conduct experiments with a simple 3-layer BN convolutional neural network (ConvNet) and a ResNet-18 on CIFAR-10 [18] and CIFAR-100 [19] datasets. Both networks are in the implementation of Lobacheva et al. [27] and we train them by optimizing cross-entropy loss using stochastic gradient descent with batch size 128 for 1000 epochs. In Appendix O, we also show that three training regimes may be observed for other neural architectures, such as multilayer perceptron and Transformer [34].

In Section 5, we analyze the training regimes of fully scale-invariant neural networks, trained on the sphere with a fixed ELR. To obtain fully scale-invariant versions of the networks we follow the same approach as in Lobacheva et al. [27]. We fix the non-scale-invariant weights, i.e., we use zero shift and unit scale in BN layers and freeze the weights of the last layer at random initialization with increased norm equal to 10. To train a network with ELR $\tilde{\eta}$, we fix the norm of the weights at the initial value $\rho$ and use the projected SGD with a fixed learning rate $\eta = \tilde{\eta}\rho^2$. We consider a wide range of ELRs: $\{10^{-k}, 2 \cdot 10^{-k}, 5 \cdot 10^{-k}\}_{k=0}^{7}$, which more than covers the values of ELR usually encountered during training of regular networks. We also used a more fine grained ELR grid closer to the boundaries between the regimes. In Section 6, we investigate how the observed training regimes transfer to more practical scenarios. We train networks in the whole parameter space with a fixed learning rate and weight decay as in Lobacheva et al. [27], Wan et al. [35]. We use weight decay of 1e-4 / 5e-4 for ConvNet / ResNet-18 in all the experiments and a range of LRs, specific for each scenario. In some experiments, we also use momentum of 0.9, standard CIFAR data augmentation, and cosine LR schedule for 200 epochs. More details can be found in Appendix C.

To make the experiments with different ELRs/LRs fully comparable, we train the networks from the same random initialization and with the same optimizer random seed. However, we show in Appendix E that the results are consistent if we vary both of them. For each visualization we choose the most representative subset of ELRs/LRs due to the difficulties of distinguishing between many noisy lines on the plots. At each training epoch, we log standard train / test statistics and the mean (over mini-batches) norm of the stochastic effective gradients for fully scale-invariant networks or the mean norm of the regular stochastic gradients in the case of standard networks. We choose this metric for the following two reasons. First, it can serve as a measure of sharpness of the minimum to which we converge in the first regime, since the mean stochastic gradient norm is strongly correlated with the trace of the gradient covariance matrix, which in turn is closely tied with the loss Hessian and the Fisher Information Matrix [13, 37, 32] (see Appendix B). Second, it represents the average (effective) step size up to multiplication by (E)LR and thus is also related to optimization dynamics.

## 5 Three regimes of training on the sphere

In this section, we investigate the three training regimes of fully scale-invariant neural networks, trained on the sphere with a fixed ELR. We first analyze the learning dynamics in each regime, including the properties of the resulting solutions and the loss landscape around them. After that, we discuss transitions between the regimes in more detail. We show the results for ConvNet on CIFAR-10, the results for other dataset-architecture pairs are consistent and presented in Appendix D.

### 5.1 Regime one: convergence

Training with the smallest ELRs results in the first training regime, which we call *the convergence regime*. As shown in Figure 1, optimization in this regime experiences a typical convergence behavior: after a number of epochs the model is able to reach regions with very low training loss and continues converging to the minimum. The speed of convergence depends on the ELR value — the higher the ELR, the faster the optimization, and the lower training loss is achieved after a fixed number of epochs. Also, training with different ELRs results in solutions with different sharpness (mean norm of stochastic effective gradients) and generalization (test error): higher ELRs lead to less sharp solution with better generalization. Similar results are well known in the literature [13, 36, 23, 30, 15, 21, 6, 3, 31, 12, 8], i.e., training neural networks using larger learning rates typically results in flatter and better generalizing optima. Furthermore, we discovered that the optima achieved after training with different ELRs not only vary in sharpness and generalization but also reside in different basins, i.e., there is a significant loss barrier on the linear interpolation between them (see Appendix I for more details).

The value of ELR affects not only the rate of convergence and the properties of the final solution; it may have an effect on the whole training trajectory as well. To analyze the training trajectories independently from the training speed, we look at the evolution of sharpness and generalization vs. training loss during the optimization. We show the corresponding plots for different ELR values of the first regime in Figure 3, left. For the lowest ELRs the trajectories coincide, but the training is too slow to converge to the low-loss region. For the remaining values, as we increase ELR, we obtain a lower trajectory, and in the low-loss region trajectories of different ELRs appear as parallel lines.

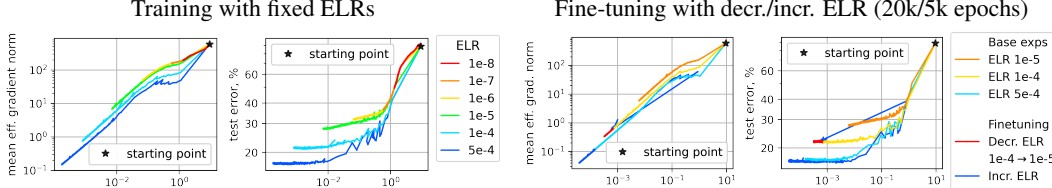

Figure 3: First training regime, ConvNet on CIFAR-10. Left: training with fixed ELRs converge to the regions with different sharpness and generalization profiles. Right: fine-tuning with decreased ELR stays on the same trajectory, while fine-tuning with increased ELR jumps out and converges to a flatter region. As baselines, we show training with the decreased/increased ELR from the initialization.

Such differences in trajectories show that training with a higher ELR not only results in a flatter final solution but also traverses less sharp regions the whole time: for any fixed value of the training loss, the point corresponding to a higher ELR has lower sharpness and better generalization. This is also consistent with recent results on the impact of higher learning rates on better conditioning of the optimization trajectory [15, 6, 12, 8].

The parallelism of the trajectories in the low-loss region is particularly interesting and leads us to the following question: do the basins to which training with different ELRs converges have different *overall* sharpness and generalization profiles? That is, can we hypothesize that all solutions reachable with SGD in a given basin have similar sharpness and generalization at the same training loss? To answer that, we take a solution achieved with a specific ELR and fine-tune it with lower and higher ELR values (see Figure 3, right). Fine-tuning with a lower ELR stays in the same basin and continues to move along the same trajectory as the pre-trained model, hence no sharper sub-minima can be found inside the flat basin. Fine-tuning with a higher ELR also maintains the same trajectory at first but jumps out of the low-loss region after a while and converges to a new basin with a profile attributed to that higher ELR. That means that the initial basin is too sharp for the higher ELR to converge. The results are consistent for different ELR values, see Appendix H; we also provide results on linear connectivity of pre-trained and fine-tuned solutions in Appendix I.

As can be seen from Figure 3, left, the range of sharpness and generalization profiles reachable with the fixed ELR training in the first regime is limited. For small ELRs, the trajectories coincide and optimization converges to the sharpest available basin. Increasing ELR results in flatter optima, but at some value the optimization stops converging to low-loss regions and demonstrates the behavior of the second training regime, which we describe next.

## 5.2 Regimes two and three: chaotic equilibrium and divergence

A further increase of ELR in the first regime leads to a sharp switch to another regime, *the chaotic equilibrium*. Figure 1 shows that after a certain threshold ELR, the optimization is no longer able to reach low-loss regions and gets stuck at some loss level. We observe that the effective gradients norm also stabilizes, and higher ELRs correspond to a higher loss but lower sharpness.

As we determined in Section 3.2, the second regime corresponds to the state when all individual effective step sizes become equal, which also leads to the stabilization of individual ELRs. In Figure 4 we depict the values (averaged over last 200 epochs) of individual ELRs and effective gradient norms w.r.t. each SI parameters group of each layer of the neural network trained with $\tilde{\eta} = 10^{-3}$. We see that the effective step sizes (ELR times effective gradient norm) indeed concentrate around a certain equilibrium value denoted as a dashed black line on the plot. In Appendix J, we provide results for other ELRs and additional plots demonstrating the stabilization of individual ELRs and effective gradient norms, which corroborates our definition of the second regime as the chaotic equilibrium.

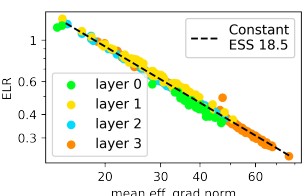

Figure 4: Equilibration of individual ESS values in the second regime. ConvNet on CIFAR-10.

In the second regime, the optimization localizes a certain region in the loss landscape and "hops along its walls". For moderate ELRs, this region is locally convex in the sense that the loss on the linear path

Fine-tuning with second regime ELRs                    Fine-tuning with first regime ELRs (2k epochs)

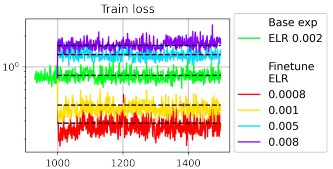 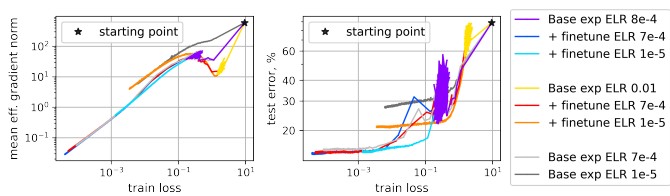

Figure 5: Second training regime, ConvNet on CIFAR-10. Left: fine-tuning with a different second regime ELR results in the training loss level corresponding to the new ELR value (such levels for all considered ELRs are depicted with dashed lines). Right: fine-tuning with first regime ELR from the low second regime ELR always results in flattest possible solutions, while from higher starting ELRs, solutions of various sharpness can be reached.

between different checkpoints from the same trajectory is convex and generally much lower than at its edges. However, for the highest ELRs, the dynamics become more chaotic and distinguished barriers may appear on the linear path. We provide the corresponding plots in Appendix K. Each ELR of the second regime determines its own loss level — the height at which the optimization "hops". Changing the ELR value during training appropriately changes the training dynamics: increasing/decreasing the ELR directly leads to the trajectory with a loss level corresponding to the new ELR value (see Figure 5, left). This significantly distinguishes the second regime from the first, where fine-tuning with a different ELR does not affect the learning trajectory, unless the new ELR is too large, in which case the optimization suddenly jumps out of the basin.

Now, what if we fine-tune a model from the second regime with a smaller ELR corresponding to the first regime? As can be seen from Figure 5, right, the resulting optima highly depend on the starting ELR (more results in Appendix L). Fine-tuning the models, pre-trained with low second regime ELRs, using any first regime ELR converges to basins with the sharpness/generalization profile attributed to the maximum ELR in the first regime. For larger second regime ELRs, fine-tuning results in a variety of trajectories depending on the chosen ELR of the first regime. That means that training with low second regime ELRs discovers regions containing only flat minima. Meanwhile, regions discovered with high ELRs contain optima of different sharpness. Moreover, fine-tuning from these regions results in the same range of sharpness/generalization profiles that was observed in the first regime.

For the highest considered ELR values, we observe the most unstable optimization behavior corresponding to the random guess accuracy (see Figure 1). We call it the third, *divergent training regime*. We explicitly compare the third regime with random walk in Appendix M.

## 5.3 Transitions between the regimes

In the end, we would like to discuss the connections between the three regimes and our global view of training on the sphere. In Figure 6, we plot the training loss for several ELR values that correspond to the three regimes and transitions between them. When the ELR is too high (purple line), the total ESS becomes so large that the model is unable to even detect any region in the parameters space with non-trivial quality, and we encounter the third regime. With a smaller ELR (light blue line), the model starts occasionally hitting the edge of the region with a (relatively) low loss but then quickly escapes it and returns back to the random guess behavior; this could be attributed to the border regime between the second and the third regimes. Setting the ELR to an even smaller value (green

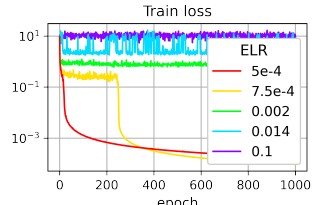

Figure 6: Three regimes and transitions between them, ConvNet on CIFAR-10.

line) results in the chaotic equilibrium, when the loss is stabilized near a certain value, which is lower the lower the ELR. For ELR values between the highest ELRs of the first regime and the lowest ones of the second regime (yellow line), we can witness a sudden change from equilibrium behavior to convergence. We conjecture that the major difference between the first and second optimization regimes and such a sharp transition between them, in particular, is due to the presence of zones of increased sharpness in the optimization trajectories of neural networks [14, 15, 21, 12, 8]. In the

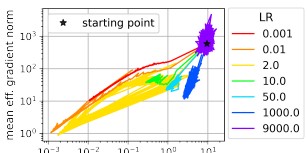
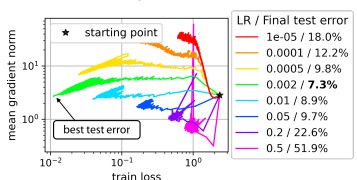
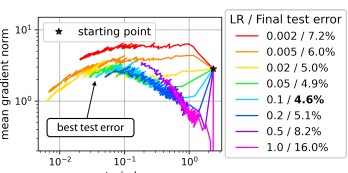

Figure 7: Training regimes in standard training. Left: training of SI networks in the whole parameter space with weight decay, ConvNet on CIFAR-10. Middle and right: conventional training with fixed LRs and cosine LR schedules, ResNet-18 on CIFAR-10. Results for other dataset-architecture pairs are presented in Appendix N.

gradient norm vs. training loss diagram in Figure 1, we can observe that the sharpness reaches its peak precisely at the transition point between the first two regimes; this leads us to a hypothesis that only training with sufficiently small ELR values allows getting through this bottleneck and enter the convergence regime. In Appendix G, we discuss the relationship between this transition and the epoch-wise double descent of the test error [28]: we discover that the sharpness peak correlates with the test error peak and also becomes much more noticeable at the presence of label noise in the training data. Finally, the smallest ELRs (orange line) correspond to a typical convergence to the optimum, i.e., the first regime, and the lower the ELR the sharper the minimum we converge to.

## 6 Regimes in standard training

In this section, we study how different training regimes of SI neural networks on the sphere are represented in more practical scenarios and what clues they can give to achieve a good minimum in terms of sharpness and generalization. First, we experiment with training SI networks in a whole parameter space, where both direction and norm dynamics influence the optimization trajectory. We train SI networks with weight decay and vary LR instead of ELR. All three regimes are also present in such training (see Figure 7, left). We argue that the difference between the first and the second regimes explains why disparate results on training dynamics are reported in the literature. Lobacheva et al. [27] have discovered that SI neural networks experience convergence with periodic jumps during optimization. We observe the same behavior for high LR values in the first regime, where the trajectories periodically jump out of the low-loss region (the yellow trajectory for LR = 2.0). On the other hand, experiments of Li et al. [26] and Wan et al. [35] with fixed LRs demonstrate that the equilibrium state is reached, which we can observe in the second training regime.

Now, let us consider conventional neural networks training, which implies optimization of all network's weights (including the non-scale-invariant ones) in the entire parameter space with momentum and data augmentation. Here we use ResNet-18 network of standard size with weight decay and try both constant and the standard cosine LR schedule (see Figure 7, middle and right). In conventional training, only the first two training regimes are present because LR also affects the non-scale-invariant parameters, and so training with very high LR values results in NaN values in the loss, since the weights of the last layer can become large enough to lead to numerical overflows. The transition between the first two regimes is not as easy to locate as in the previous experiments. All the training trajectories end up in regions with relatively high training loss, mainly because of the data augmentation, which makes the task much harder to learn and requires more time to converge to the low-loss region. Moreover, the task may become too hard for some models to achieve a low training loss at all. We show in Appendix F, that similar results are observed in case of training SI networks that are too small for the task in hand.

In the experiments with constant LRs, training with lower LRs ($< 0.01$) passes the region with the highest sharpness along the trajectory and starts converging to a flatter region, while training with higher LRs ($\geq 0.01$) gets stuck in the high-sharpness zone. The former can be attributed to the first training regime, and the latter to the second one. Note that, in accordance with Section 5, the best test accuracy is achieved with the largest LR of the first regime.

In the experiments with cosine LR schedule, we decrease LR from some starting LR value to zero during 200 epochs of training. If we start with a low LR ($\leq 0.02$), optimization is carried out all or

almost all the time with LRs of the first regime, and the first regime trajectory is observed. Moreover, for higher starting LRs, the solution with better sharpness profile and generalization is achieved. If we start with a high LR ($\geq 0.5$), then most of the training is done with LRs of the second regime, and optimization does not have enough time with low LR values to converge to the low-loss region. The medium starting LR values are of the most interest here (between $0.02$ and $0.5$). In this case, training first has substantial time with gradually decreasing LRs of the second regime, which helps it to localize the region with the flattest optima, and then has just enough time with the first regime LRs to pass the high-sharpness zone and converge to the minimum. With such training, solutions with better sharpening/generalization profiles are achieved, which is consistent with the results of Section 5.2 and the existing literature on pre-training with high LR values [23, 11].

## 7    Conclusion and Discussion

In this work, we investigated the properties of training scale-invariant neural networks on the sphere using projected SGD. This method allowed us to study the intrinsic domain of SI models more accurately. We discovered three regimes of such training depending on the effective learning rate value: convergence, chaotic equilibrium, and divergence. They uncover multiple properties of the intrinsic loss landscape of scale-invariant neural networks: some are related to diverse previous results in deep learning, and some appear as novel. We showed that these regimes are present in conventional neural network training as well and can provide intuition on how to achieve better optima.

**Possible future directions**   Speaking about the potential prospects for further research, three directions can be distinguished. First, it would be interesting to develop more solid and general theoretical groundings for the observed phenomena. For instance, a closer look at the negative feedback principle in eq. (5) could potentially shed more light on the ESS/ELR dynamics in the first and second regimes compared to the current decay vs. alignment dichotomy and maybe better explain occasional transitions from the second to the first regime for certain ELRs in Section 5.3. Second, the analysis could be extended to other tasks, architectures, and especially optimizers like Adam [16], which can be challenging due to their indirect impact on the effective direction and effective learning rate in SI models [29]. Third, it is very tempting to investigate other potential practical implications of the three regimes for conventional training. The analysis of various LR schedules from the perspective of the discovered training regimes is intriguing and might help in designing more efficient and more explainable LR schedules. For example, from our experiments we observe that decreasing (E)LR in the first regime during training does not influence the final solution but only slows down the convergence; at the same time, for the optimal final performance, optimization should run through the low second regime (E)LRs, which allows to achieve basins containing solutions with the best sharpness/generalization properties.

**Limitations and societal impact**   The main limitations of our work are applying a single optimization method on the sphere (projected SGD), experimenting only with BN networks as SI models, and considering only one of the symmetries that may conceal the actual properties of neural networks intrinsic domain [20]. To our knowledge, our work does not have any negative societal impact.

## Acknowledgments and Disclosure of Funding

We would like to thank Mikhail Burtsev for the valuable discussions. The publication was supported by the grant for research centers in the field of AI provided by the Analytical Center for the Government of the Russian Federation (ACRF) in accordance with the agreement on the provision of subsidies (identifier of the agreement 000000D730321P5Q0002) and the agreement with HSE University №70-2021-00139. The empirical results were supported in part through the computational resources of HPC facilities at HSE University [17]. Additional revenues of the authors for the last three years: laboratory sponsorship by Samsung Research, Samsung Electronics and Huawei Technologies; honorarium for teaching by MSU; internship at Yandex.

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
