# A Theory

In this section, we provide proofs and additional details for Section 3.

## A.1 Norm constraint: total vs. individual

In this section, we will give additional arguments in favor of our choice of constraining the total norm, rather than each individual SI group. Consider a more general formulation than (1), where we relax the condition on fixing the total norm strictly at $\rho$:

$$\begin{cases} \min_{\boldsymbol{\theta} \in \mathbb{R}^P} F(\boldsymbol{\theta}) \\ \|\boldsymbol{\theta}\|^2 \leq \rho^2. \end{cases} \tag{10}$$

Arora et al. [1] show that after a gradient step, the norm of SI parameters can only increase. That means that if we solved the problem (10) using the projected gradient descent method starting from $\boldsymbol{\theta}^{(0)} : \left\|\boldsymbol{\theta}^{(0)}\right\| = \rho$, we would end up with the same algorithm, as presented in eq. (2), because we would project $\boldsymbol{\theta}$ onto $\mathcal{S}(\rho)$ at each iteration. Thus, this relaxation does not any significantly vary the setup considered in the main text.

The Lagrangian function associated with (10) is

$$\mathcal{L}(\boldsymbol{\theta}, \lambda) = F(\boldsymbol{\theta}) + \lambda \|\boldsymbol{\theta}\|^2 - \lambda \rho^2, \ \lambda \geq 0. \tag{11}$$

Minimizing (11) w.r.t. $\boldsymbol{\theta}$ is equivalent to standard training with $L_2$ regularizer or, in the case of gradient descent, applying weight decay with coefficient $\lambda$ in the optimizer. This is a perfectly normal procedure when training machine learning models.

If we, however, tried putting constraints on each individual SI group norm, we would end up with the following optimization problem:

$$\begin{cases} F(\boldsymbol{\theta}) \to \min_{\boldsymbol{\theta} \in \mathbb{R}^P} \\ \|\boldsymbol{\theta}_i\|^2 \leq \rho_i^2, \ i = 1, \ldots, n, \end{cases} \tag{12}$$

which yields the following Lagrangian function:

$$\mathcal{L}(\boldsymbol{\theta}, \lambda_1, \ldots, \lambda_n) = F(\boldsymbol{\theta}) + \sum_{i=1}^n \lambda_i \|\boldsymbol{\theta}_i\|^2 - \sum_{i=1}^n \lambda_i \rho_i^2, \ \lambda_i \geq 0, \ i = 1, \ldots, n. \tag{13}$$

Now, minimizing (13) w.r.t. $\boldsymbol{\theta}$ would imply setting $n$ separate weight decay coefficients $\lambda_i$ (one for each SI group), which has much less to do with actual practice.

## A.2 Derivations for Section 3.1

We begin with a formal derivation of the formulas in Section 3.1. We remind that we consider a function $F(\boldsymbol{\theta})$ whose parameters can be split into $n$ SI groups: $\boldsymbol{\theta} = (\boldsymbol{\theta}_1, \ldots, \boldsymbol{\theta}_n)$. We solve an optimization problem (1) with projected gradient descent (2). For each SI group $\boldsymbol{\theta}_i$ we denote its norm as $\rho_i \equiv \|\boldsymbol{\theta}_i\|$, its effective gradient norm as $\tilde{g}_i = g_i \rho_i$, where $g_i \equiv \|\nabla_{\boldsymbol{\theta}_i} F(\boldsymbol{\theta})\|$, and its ELR as $\tilde{\eta}_i = \eta / \rho_i^2$. Similarly, for the whole parameters vector $\boldsymbol{\theta}$ we define $\tilde{g} = g\rho$, where $g \equiv \|\nabla_{\boldsymbol{\theta}} F(\boldsymbol{\theta})\|$, and $\tilde{\eta} = \eta / \rho^2$.

Equation (3) can be directly obtained from the basic constraint on the total parameters norm:

$$\sum_{i=1}^n \rho_i^2 = \rho^2 \implies \sum_{i=1}^n \frac{1}{\tilde{\eta}_i} = \sum_{i=1}^n \frac{\rho_i^2}{\eta} = \frac{\rho^2}{\eta} = \frac{1}{\tilde{\eta}}. \tag{14}$$

Equation (4) expresses the total squared ESS value as a weighted average of squared individual ESS values and can be obtained the following way:

$$(\tilde{\eta}\tilde{g})^2 = \tilde{\eta}\eta g^2 = \{g^2 = \sum_{i=1}^n g_i^2\} = \tilde{\eta} \sum_{i=1}^n \eta g_i^2 = \tilde{\eta} \sum_{i=1}^n \tilde{\eta}_i \tilde{g}_i^2 = \sum_{i=1}^n \omega_i (\tilde{\eta}_i \tilde{g}_i)^2, \ \omega_i \equiv \frac{\tilde{\eta}}{\tilde{\eta}_i}. \tag{15}$$

Note that from eq. (3) it follows that $\sum_{i=1}^{n} \omega_i = 1$, i.e., the rightmost expression of eq. (15) is indeed a convex combination of of squared individual ESS values with weights $\omega_i \propto \frac{1}{\tilde{\eta}_i}$.

Finally, the update rule for ELRs (5) in process (2) can be obtained from a similar expression for the parameters norm updates:

$$(\rho_i^{(t+1)})^2 = \frac{\left\| \boldsymbol{\theta}_i^{(t)} + \eta \nabla_{\boldsymbol{\theta}_i} F(\boldsymbol{\theta})^{(t)} \right\|^2}{\left\| \boldsymbol{\theta}^{(t)} + \eta \nabla_{\boldsymbol{\theta}} F(\boldsymbol{\theta})^{(t)} \right\|^2} \rho^2 = \{\langle \boldsymbol{\theta}_i, \nabla_{\boldsymbol{\theta}_i} F(\boldsymbol{\theta}) \rangle = 0 \, [24]\} =$$

$$= \frac{(\rho_i^{(t)})^2 + \eta^2 g_i^2}{1 + (\tilde{\eta} \tilde{g}^{(t)})^2} = (\rho_i^{(t)})^2 \frac{1 + (\tilde{\eta}_i^{(t)} \tilde{g}_i^{(t)})^2}{1 + (\tilde{\eta} \tilde{g}^{(t)})^2}. \quad (16)$$

### A.3 Proof of Proposition 1

Now, we accurately formulate and prove Proposition 1.

**Proposition 2** *Consider a SI function $f_\alpha(x,y) = \alpha \frac{x^2}{x^2+y^2}$. Let it be optimized using projected gradient descent (2) with learning rate $\eta \equiv \tilde{\eta}$ on the unit sphere starting from a point $(x^{(0)}, y^{(0)})$ such that $x^{(0)} \neq 0$ and $y^{(0)} \neq 0$. Then the following results hold:*

1. *$\tilde{\eta} < \frac{1}{\alpha}$ is a sufficient condition for linear convergence of the function to zero (global minimum);*

2. *$\tilde{\eta} > \frac{1}{\alpha}$ is a sufficient condition for the function to stabilize at value $\frac{1}{2}\left(\alpha - \frac{1}{\tilde{\eta}}\right)$.*

**Remark 1** *If $\tilde{\eta} = \frac{1}{\alpha}$, the convergence to the minimum is still preserved, although it may not be as stable, so we omit this case as degenerate.*

**Remark 2** *The above formulation allegedly lacks the third (divergent) regime. We draw the line between the second and the third regimes by comparing them with random guess quality (see Appendix M): the third regime is much more reminiscent of the random walk behavior than the second one (chaos $\gg$ equilibrium). Using the same criterion, we may notice that in the limit of $\tilde{\eta} \to \infty$ the value of the function $f_\alpha(x,y)$ around which optimization "stabilizes" is $\frac{\alpha}{2}$ corresponding to the expected value of the function given that input points are randomly sampled on the sphere, which totally accords with our intuition.*

**Proof.** To study the convergence properties of $f_\alpha(x,y)$, note first that it depends only on the $x/y$ ratio, which we denote as $r \equiv x/y$:

$$f_\alpha(x,y) = \alpha \frac{x^2}{x^2+y^2} = \alpha \frac{r^2}{1+r^2}. \quad (17)$$

That allows us to conclude that $f(x^{(t)}, y^{(t)}) \xrightarrow[t\to\infty]{} 0 \iff r^{(t)} \xrightarrow[t\to\infty]{} 0$, thus studying the convergence of $f_\alpha$ to the minimum is equivalent to studying the convergence of $r$ to zero. The gradient of the function is

$$\nabla f_\alpha(x,y) = \frac{2\alpha xy}{(x^2+y^2)^2}[y, -x]^T. \quad (18)$$

That induces the following update rule for $r$ (note that $(x^{(t)})^2 + (y^{(t)})^2 = 1$):

$$r^{(t+1)} = \frac{x^{(t+1)}}{y^{(t+1)}} = \frac{x^{(t)} - 2\alpha\tilde{\eta} x^{(t)}(y^{(t)})^2}{y^{(t)} + 2\alpha\tilde{\eta}(x^{(t)})^2 y^{(t)}} = \frac{x^{(t)}}{y^{(t)}}\left(1 - \frac{2\alpha\tilde{\eta}}{1 + 2\alpha\tilde{\eta}(x^{(t)})^2}\right) =$$

$$= r^{(t)}\left(1 - \frac{2\alpha\tilde{\eta}}{1 + 2\alpha\tilde{\eta}(r^{(t)})^2/(1+(r^{(t)})^2)}\right). \quad (19)$$

Ultimately, we get the following expression for the next absolute value of $r$:

$$|r^{(t+1)}| = \kappa^{(t)}|r^{(t)}|, \quad \kappa^{(t)} \equiv \left| 1 - \frac{2\alpha\tilde{\eta}}{1 + 2\alpha\tilde{\eta}(r^{(t)})^2/(1+(r^{(t)})^2)} \right|. \quad (20)$$

Now, we prove the first statement. If $\tilde{\eta} < \frac{1}{\alpha}$, then

$$0 < \frac{2\alpha\tilde{\eta}}{1 + 2\alpha\tilde{\eta}} < \frac{2\alpha\tilde{\eta}}{1 + 2\alpha\tilde{\eta}(r^{(t)})^2/(1 + (r^{(t)})^2)} < 2\alpha\tilde{\eta} < 2, \tag{21}$$

from which we obtain that $\kappa^{(t)} < 1$ in eq. (20), which implies linear convergence of $r$ to zero.

For the second statement, based on eq. (21), one can obtain that

$$\kappa^{(t)} > 1 \iff \frac{2\alpha\tilde{\eta}}{1 + 2\alpha\tilde{\eta}(r^{(t)})^2/(1 + (r^{(t)})^2)} > 2 \iff \frac{(r^{(t)})^2}{(1 + (r^{(t)})^2)} < \frac{\alpha\tilde{\eta} - 1}{2\alpha\tilde{\eta}} \iff$$

$$\iff (r^{(t)})^2 < (r^*)^2 \equiv \frac{\alpha\tilde{\eta} - 1}{\alpha\tilde{\eta} + 1}. \tag{22}$$

That essentially means that $r^2$ must stabilize at level $(r^*)^2$ since otherwise, if $(r^{(t)})^2 < (r^*)^2$, $\kappa^{(t)} > 1$ and the sequence begins to increase, and if $(r^{(t)})^2 > (r^*)^2$, $\kappa^{(t)} < 1$ and the sequence is decreasing. Finally, note that the level $(r^*)^2$ from eq. (22) corresponds to the function's value $\frac{1}{2}\left(\alpha - \frac{1}{\tilde{\eta}}\right)$ due to eq. (17). $\blacksquare$

## A.4 More formally on the results of Section 3.2

In this section, we provide a more formal argument on the results of Section 3.2.

Minimization of function (7) on the unit sphere can be represented as a separable optimization problem with a single uniform constraint on the parameters norm:

$$\begin{cases} F(\boldsymbol{x}, \boldsymbol{y}) \to \min_{\boldsymbol{x},\boldsymbol{y}} \\ \|\boldsymbol{x}\|^2 + \|\boldsymbol{y}\|^2 = 1 \end{cases} \iff \begin{cases} \alpha_i f(x_i, y_i) \to \min_{x_i, y_i}, \ i = 1, \dots, n \\ \sum_{i=1}^n x_i^2 + y_i^2 = 1. \end{cases} \tag{23}$$

According to the results of Section 3.1, solving it with the projected gradient method (2) with a fixed total ELR $\tilde{\eta}$ would be similar to running $n$ projected gradient methods for each subfunction $\alpha_i f(x_i, y_i)$ with varying individual ELRs $\tilde{\eta}_i$, related by eq. (3). Proposition 1 states that for each subfunction to converge, its individual ELR $\tilde{\eta}_i$ must remain below the $\frac{1}{\alpha_i}$ threshold or, equivalently,

$$\frac{1}{\tilde{\eta}_i} > \alpha_i. \tag{24}$$

If the first regime condition (8) is fulfilled, then $\frac{1}{\tilde{\eta}} = \sum_{i=1}^n \frac{1}{\tilde{\eta}_i} > \sum_{i=1}^n \alpha_i$, and thus there is enough capacity to satisfy condition (24) for each subfunction and successfully converge to the minimum.

If, conversely, $\tilde{\eta} > \frac{1}{\sum_{i=1}^n \alpha_i}$, then at each iteration at least one of the individual ELRs exceeds its convergence threshold: $\tilde{\eta}_i > \frac{1}{\alpha_i}$. Due to Proposition 1, that subfunction's value will tend to stabilize at $\frac{1}{2}\left(\alpha_i - \frac{1}{\tilde{\eta}_i}\right)$. Generalizing that argument, at any moment each subfunction with individual ELR $\tilde{\eta}_i$ tends to $\max\left\{0, \frac{1}{2}\left(\alpha_i - \frac{1}{\tilde{\eta}_i}\right)\right\}$, and thus the whole function $F(\boldsymbol{x}, \boldsymbol{y})$ tends to their sum:

$$\sum_{i=1}^n \max\left\{0, \frac{1}{2}\left(\alpha_i - \frac{1}{\tilde{\eta}_i}\right)\right\} \geq \frac{1}{2}\sum_{i=1}^n \left(\alpha_i - \frac{1}{\tilde{\eta}_i}\right) = \frac{1}{2}\left(\sum_{i=1}^n \alpha_i - \frac{1}{\tilde{\eta}}\right) > 0. \tag{25}$$

That means that in the second regime the function value is pushed away from zero towards some positive value at every iteration. The same applies to its gradient, from what we conclude that the total effective step size is also nonvanishing. That implies that the dynamics (5) work in the "undamped" regime, i.e., the squared effective step size values in the expression cannot be neglected. As stated in the main text, the only way for such dynamics to stabilize is to equalize the ESS values.

In Proposition 1, we prove that once ELR $\tilde{\eta}$ is larger than the $\frac{1}{\alpha}$ threshold, the function $f_\alpha(x, y)$ is stabilized at $\frac{1}{2}\left(\alpha - \frac{1}{\tilde{\eta}}\right)$. In the proof (see Proposition 2), we show that it follows from the stabilization of the $r^2 \equiv x^2/y^2$ ratio at $(r^*)^2 \equiv \frac{\alpha\tilde{\eta}-1}{\alpha\tilde{\eta}+1}$. Using eq. (18), it can be shown that the squared effective gradient norm of $f_\alpha(x, y)$ equals

$$\tilde{g}_\alpha^2 \equiv \|\nabla f_\alpha(x, y)\|^2 (x^2 + y^2) = 4\alpha^2 \frac{x^2 y^2}{(x^2 + y^2)^2} = 4\alpha^2 \frac{r^2}{(1 + r^2)^2}. \tag{26}$$

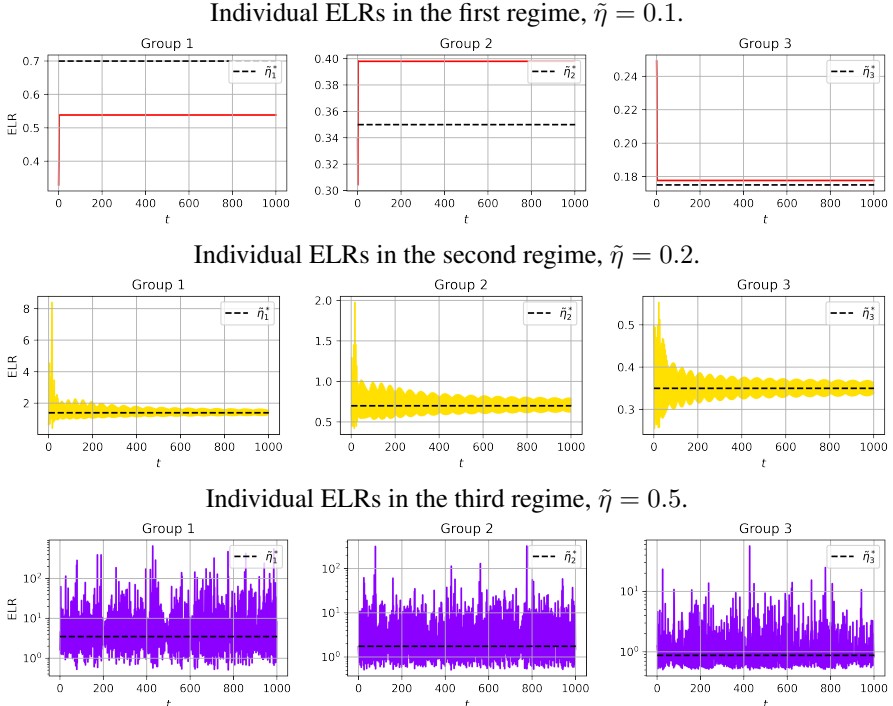

Figure 8: Individual ELRs from the experiment at the end of Section 3.2 in three regimes.

Now, substituting $(r^*)^2$ for $r^2$ in eq. (26) results in $\alpha^2 - \frac{1}{\tilde{\eta}^2}$, and thus the corresponding squared ESS value is $\alpha^2 \tilde{\eta}^2 - 1$. In the chaotic equilibrium state, when all the individual ESS values are equal, we obtain the following system of equations for individual ELRs:

$$\begin{cases} \alpha_i^2 \tilde{\eta}_i^2 - 1 = \alpha_j^2 \tilde{\eta}_j^2 - 1, \ i \neq j \\ \sum_{i=1}^n \frac{1}{\tilde{\eta}_i} = \frac{1}{\tilde{\eta}}. \end{cases} \tag{27}$$

Eq. (9) solves the above system and hence represents the ideal equilibrium values that stabilize the dynamics (5) for the function $F(\boldsymbol{x}, \boldsymbol{y})$. Indeed, we verify in the experiment that individual ELRs in average equal (9) in the second regime (see below).

### A.5 Additional plots for the example in Section 3.2

Here we provide additional plots depicting the behavior of individual ELRs in the toy example at the end of Section 3.2. Figure 8 illustrates the plots. Each row corresponds to a certain total ELR value (regime), each column corresponds to a certain SI parameters group. We can see that in the first regime, individual ELRs stabilize long before reaching their equilibrium limit (9) since ESS values quickly decay to zero and dynamics (5) begin to fade. In the second regime, individual ELR values closely match (in average by iterations) their corresponding equilibrium values, as the theory predicts. Finally, the third regime shows very chaotic behavior and no stabilization is visible.

## B Sharpness measures

In our experiments, we use the averaged over mini-batches norm of the (effective) stochastic gradient as a measure of both local loss sharpness and optimization step length. While the second application appears obvious due to the nature of gradient optimization, we would like to provide more details on the first point.

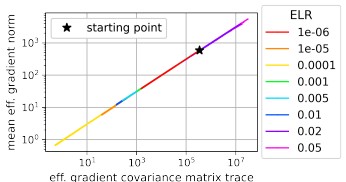

Figure 9: Mean gradient norm (28) vs. gradient covariance trace (29) during training. The two measures very strongly correlate. ConvNet, CIFAR-10.

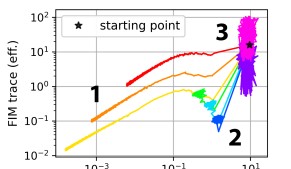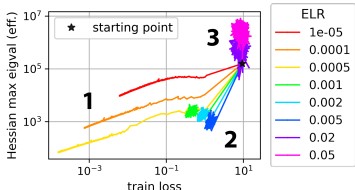

Figure 10: Training loss vs. sharpness measure diagrams with respect to different sharpness measures: Fisher Information Matrix trace (left) and maximum eigenvalue of Hessian (right). ConvNet on CIFAR-10.

Basically, we measure the following value:[3]

$$\mathbb{E}_{\mathcal{B}} \left\| \nabla L_{\mathcal{B}}(\boldsymbol{\theta}) \right\|, \tag{28}$$

where $L_{\mathcal{B}}(\boldsymbol{\theta})$ denotes the loss on the mini-batch $\mathcal{B}$ and expectation is taken over the mini-batches, and we additionally multiply it by the parameters norm $\|\boldsymbol{\theta}\|$ in the effective case. Expression (28) resembles the formula of the stochastic gradient covariance matrix trace:

$$\text{Tr}\left(\mathbb{E}_{\mathcal{B}}\nabla L_{\mathcal{B}}(\boldsymbol{\theta})\nabla L_{\mathcal{B}}(\boldsymbol{\theta})^T\right) = \mathbb{E}_{\mathcal{B}}\left\|\nabla L_{\mathcal{B}}(\boldsymbol{\theta})\right\|^2. \tag{29}$$

The covariance matrix is known to be correlated with the loss Hessian matrix and the Fisher Information Matrix [32], and thus its trace can be regarded as a universal and easy-to-calculate measure of sharpness. Note that eq. (28) and eq. (29) are identical up to taking the square of the norm under expectation. Empirically, we find that these measures are highly correlated, which could be explained by a low variance of gradient norms across mini-batches (see Figure 9). We also consider other common sharpness measures and provide training loss vs. effective Fisher Information Matrix (FIM) trace and training loss vs. effective Hessian largest eigenvalue diagrams in Figure 10 that demonstrate the same pattern as in our main diagram in Figure 1.

## C  Experimental details

**Datasets and architectures.** We conduct experiments with a simple 3-layer BN convolutional neural network (ConvNet) and a ResNet-18, on CIFAR-10 [18] and CIFAR-100 [19] datasets. We use the implementation of both architectures available at `https://github.com/tipt0p/periodic_behavior_bn_wd`. In this implementation, additional BN layers are inserted in Resnet-18 to make the majority of neural network weights scale-invariant. CIFAR datasets are distributed under the MIT license, and the code is under Apache-2.0 License.

We use the standard PyTorch initialization for all layers. ConvNet contains an input convolutional layer with $k$ filters followed by BN and ReLU non-linearity, then three blocks of convolution + BN + ReLU + maxpool with $2k/4k/8k$ filters in convolutional layers, respectively, and then maxpool + output linear layer. For ConvNet, we use width factor $k = 32$ for all experiments on CIFAR-10 (except Appendix F, where we vary network width), and $k = 64$ for all experiments on CIFAR-100 (the dataset is too complex for ConvNet with width factor 32 to reach low training loss and clearly depict the first training regime). In the case of SI ResNet (Appendix D and Appendix N), we use width factor of 32 for both datasets, while for conventional training (Section 6 and Appendix N) we use ResNet of standard width (width factor 64).

Most of the experiments are conducted with the scale-invariant modifications of both architectures obtained using the approach of Lobacheva et al. [27]. All non-scale-invariant weights, i.e., BN affine parameters and the last layer's parameters, are fixed. For BN layers, zero shift and unit scale parameters are used. The bias vector of the last layer is fixed at random initialization, while the weight matrix is fixed at rescaled random initialization (its norm equals 10).

**Training.** We train all networks using SGD with a batch size of 128. For training in the whole parameter space we use weight decay of 1e-4/5e-4 for ConvNet/ResNet-18. In the experiments with

---

[3]In the effective case, we measure this and other metrics on the unit sphere.

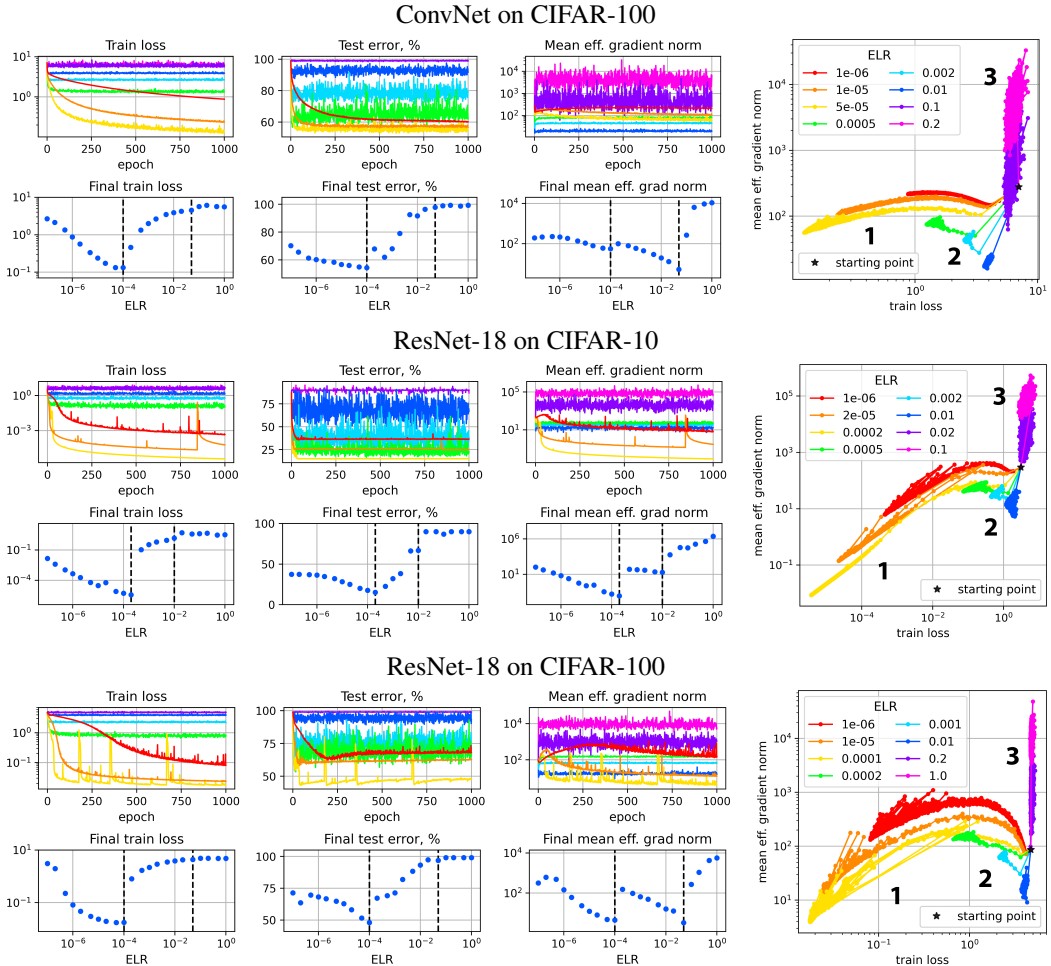

Figure 11: Three regimes of SI neural network training on the sphere. (1) convergence for the lowest ELRs, (2) chaotic equilibrium for medium ELRs, and (3) divergence for the highest ELRs. Dashed lines on scatter plots denote borders between the regimes.

momentum, we use the momentum of 0.9, in experiments with cosine LR schedule we train models for 200 epochs. In the experiments with data augmentation, we use standard CIFAR augmentations: random crop (size: 32, padding: 4) and random horizontal flip. All models were trained on NVidia Tesla V100 or NVidia GeForce GTX 1080. Obtaining the results reported in the paper took approximately 10K GPU hours. In all experiments we log all metrics after each epoch, computing train loss and its gradients by making an additional pass through the training dataset.

## D Three regimes of training on the sphere

In this section, we provide the overview plots for other dataset-architecture pairs complementary to Figure 1 in the main text (see Figure 11). Three training regimes are present for all dataset-architecture pairs. Also, we observe periodic behavior of training dynamics in some ResNet-18 plots, even though we train our models on the fixed sphere. This is very reminiscent of the results of Lobacheva et al. [27], as they also observed periodic behavior when trained SI models in the whole parameters space using weight decay. Lobacheva et al. [27] showed that these sudden jumps in training loss are due to the norm of the scale-invariant parameters becoming too low. We hypothesize that similar reasons may cause periodic behavior in our case, since in relatively complex models such as ResNet, some groups of SI parameters may also come too close to the origin even at a fixed total norm, resulting in an explosion of gradients with respect to those groups.

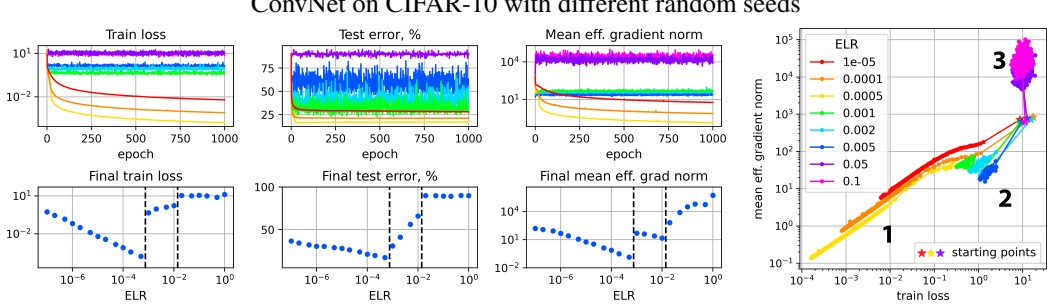

Figure 12: Three regimes of SI neural network training on the sphere with different random seed for each run (each ELR). The figure reproduces Figure 1 in the main text, including axes and the range of ELRs.

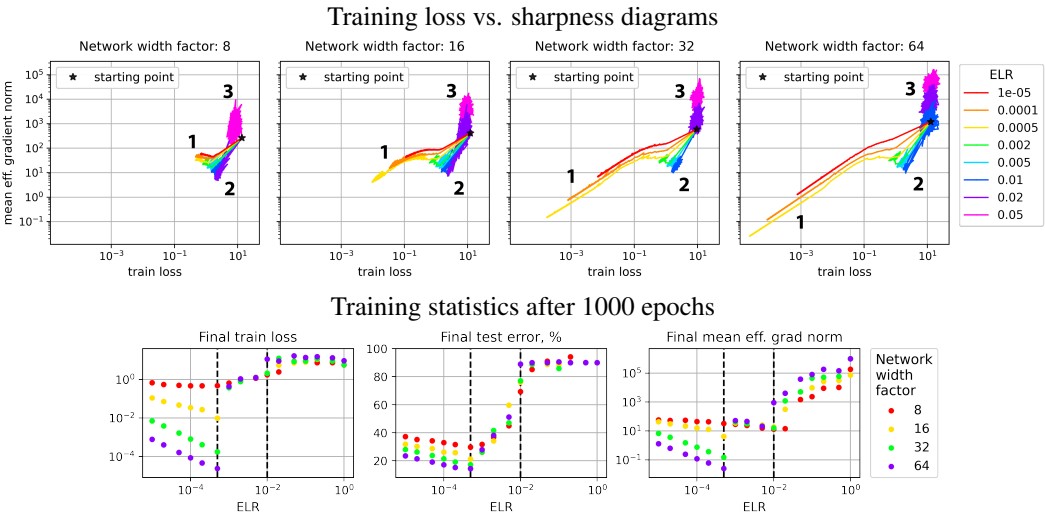

Figure 13: Three training regimes on the sphere for SI neural networks of different width. ConvNet on CIFAR-10. Axes limits are the same in each plot of the first line for convenient comparison. Dashed lines on scatter plots denote borders between the regimes for the networks of width factor 32, that we use in the rest of the paper.

# E  Three regimes: different random seeds

In this appendix, we provide plots analogous to Figure 1 in the main text but when each experiment is run with its own random seed, i.e., different runs (different ELRs) have different random initialization and mini-batch order during optimization. We report the results in Figure 12. It can be seen that the illustration is very similar to Figure 1, thus our findings withstand the randomness of optimization.

# F  Three regimes: different network sizes and data augmentation

In this section, we provide additional experiments about the dependence of the three regimes manifestation on the network size and data complexity.

In Figure 13, we demonstrate the diagrams of training loss vs. sharpness along with final training statistics, when we vary the network size (ConvNet of different width on CIFAR-10). We see that the difference between the regimes becomes much more noticeable for larger models. The smallest models (width factor 8) may not even show clear signs of transition to the first regime, i.e., convergence, and the border between the first two regimes becomes vague (see the scatter plots for the training loss and gradients). We can also notice that the second regime appears very similar for

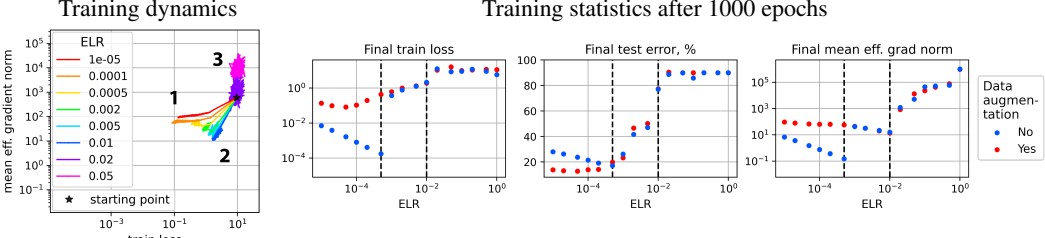

Figure 14: Three training regimes on the sphere for SI neural networks with data augmentation. ConvNet on CIFAR-10. Axes limits are the same as Figure 13 for convenient comparison. Dashed lines on scatter plots denote borders between the regimes for training without data augmentation, that we use in the rest of the paper.

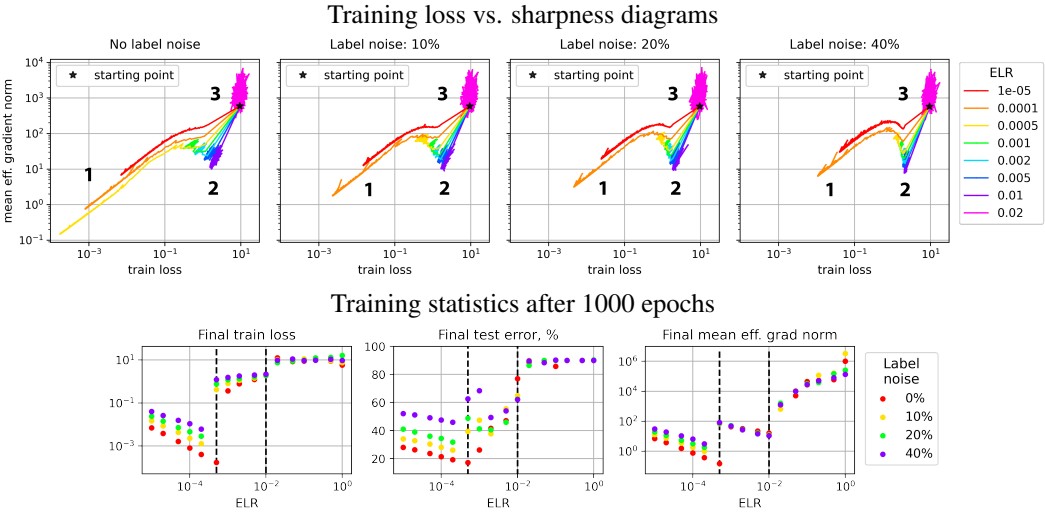

Figure 15: Three training regimes on the sphere for SI neural networks on data with different levels of label noise. ConvNet on CIFAR-10. Axes limits are the same in each plot of the first line for convenient comparison. Dashed lines on scatter plots denote borders between the regimes for training without label noise, that we use in the rest of the paper.

all network sizes, especially on the scatter plots, however, the transition between the second and third regimes happens at lower ELRs for wider networks.

Figure 14 shows the corresponding plots for ConvNet on CIFAR-10 with data augmentation. In scatter plots we provide the results for the same dataset-architecture pair without data augmentation for comparison. We observe that making dataset more complex, e.g., by turning data augmentation on, leads to a similar behavior in terms of the training loss and gradients norm as using a smaller neural network on the original dataset.

## G    Transition between regimes 1/2 and the double descent

Nakkiran et al. [28] discovered that deep learning models, especially when trained with noisy labels, can undergo the double descent behavior of the test error during training (epoch-wise Double Descent), and the effect becomes more noticeable the more noise is added to the training data. We replicate this setup and repeat our experiments with added label noise, i.e., we randomly select a portion of training objects and independently set their labels to the noise sampled from the uniform distribution over the classes. Figure 15 demonstrates the results for varying portion of corrupted labels. Note that the more noise is added to the training data, the clearer the sharpness peak becomes at the transition point between the first two regimes.

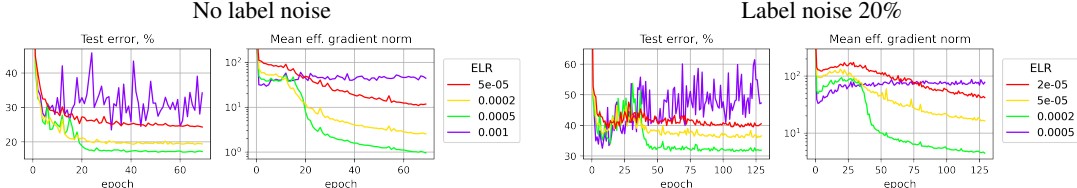

Figure 16: The first epochs of training without label noise (left) and with 20% label noise (right). Double descent behavior is observed both in test error and sharpness. ConvNet on CIFAR-10.

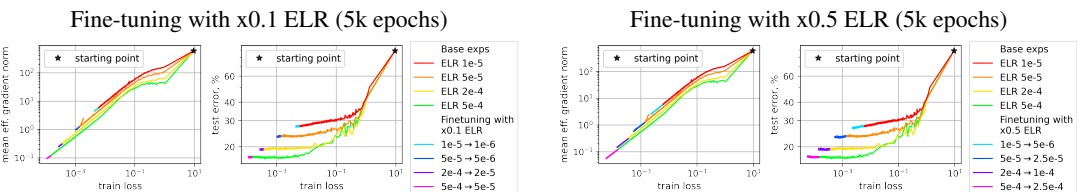

Figure 17: First training regime. Fine-tuning with decreased ELR maintains the same trajectory regardless of the initial and final ELR. ConvNet on CIFAR-10.

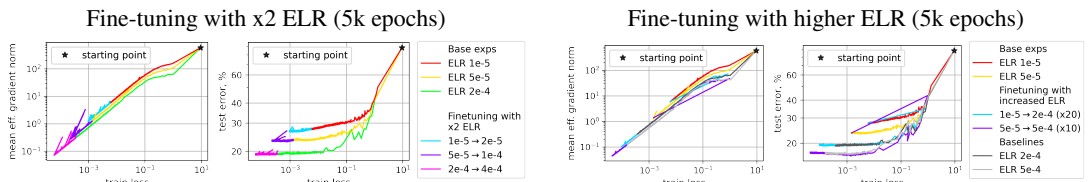

Figure 18: First training regime. Fine-tuning with a slightly increased ELR maintains the same trajectory, while with a substantially increased ELR, the training destabilizes and converges to a flatter region. ConvNet on CIFAR-10.

In Figure 16, we additionally plot the first epochs of training without label noise (left) and with 20% label noise (right). With label noise, we can clearly see that in the first regime both test error and sharpness demonstrate the double descent behavior and peak approximately at the same time. Without label noise, these peaks are not that apparent, but they also exist. At higher ELR values in the first regime, the double descent becomes more distinct, but as soon as we enter the second regime, the second descent disappears and the model gets stuck at the peak.

## H    First regime: fine-tuning with increased or decreased ELR

In this section, we provide additional results with decreased/increased ELR fine-tuning in the first regime. In Figure 17, we consider a wider range of initial ELR values when decreasing ELR. Sharpness and generalization profiles always maintain the same trajectory confirming that SGD cannot reach a sharper region after converging to the basin in the first regime.

Figure 18, left shows that basins in the first regime are also tolerant to fine-tuning with slightly increased ELR, except for the fact that the trajectories become more noisy. A greater increase in ELR (Figure 18, right) leads to destabilization and subsequent convergence to a new flatter region with sharpness/generalization profile corresponding to that of the final ELR value.

## I    First regime: linear mode connectivity

In this appendix, we provide additional plots on the linear mode connectivity in the first regime.

The optima achieved after learning with different ELRs not only vary in sharpness and generalization but also reside in different basins even if training starts from the same initialization and uses the

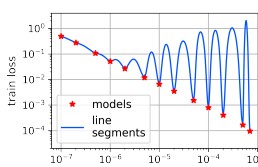
Training with constant ELR

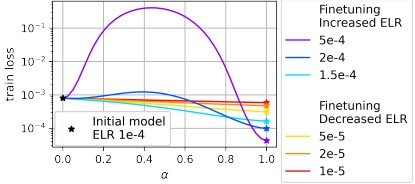
Fine-tuning with increased/decreased ELR

Figure 19: First training regime. Left: solutions achieved with different ELRs are not linearly connected. Right: the pre-trained and fine-tuned solutions are linearly connected for lower and moderately higher ELRs and not for significantly larger ELRs. ConvNet on CIFAR-10.

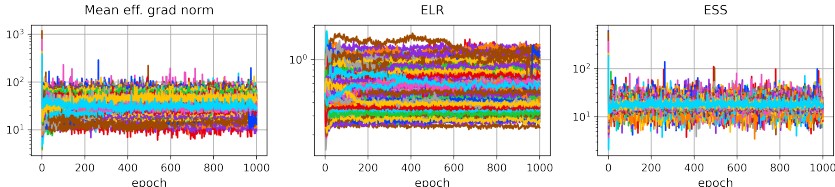

Figure 20: Individual mean effective gradient norms, ELRs, and ESS values of all SI parameters groups in the model by iterations. Stabilization of all metrics is observed; ESS values tend to cluster around the same value. ConvNet on CIFAR-10; total ELR is 0.001.

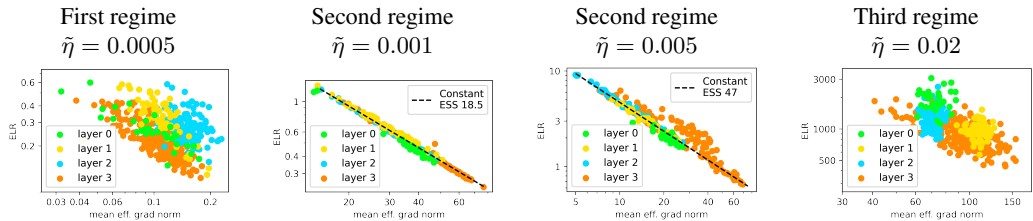

Figure 21: Individual ELRs and mean effective gradient norms of all SI parameters groups in the model (averaged over the last 200 epochs) for different total ELR values. Individual ESS equilibration is distinct in the second regime (note the dashed line). ConvNet on CIFAR-10.

same order of batches. In Figure 19, left, we check the linear connectivity of solutions achieved with different ELRs in the first regime. For each pair of neighboring ELRs, we connect the corresponding final weights with a linear path and plot the training loss along it. The results show that the solutions that converged to a low-loss region indeed lie in different basins, as there exist barriers of high loss between them.

Fine-tuning with a lower ELR does not change the basin sharpness/generalization profile, while a larger ELR may result in jumping out and converging to a new basin with its own profile. In Figure 19, right, we provide results on linear connectivity of pre-trained and fine-tuned solutions in the first regime. We can notice that the training loss is monotonically decreasing along a linear path connecting the pre-trained and the fine-tuned solution for lower and moderately higher ELR values. In contrast, for significantly larger ELRs, the linear segment has a noticeable barrier; moreover, we observe it only for the ELRs that lead to a jump in the training trajectory. This confirms our intuition that decreasing ELR maintains the same basin, while increasing ELR may result in a different one.

## J Second regime: equilibration

In this section, we provide additional plots for Figure 5, left in the main text. In Figure 20 we give additional plots demonstrating that individual metrics (effective gradient norms, ELRs, and ESS values) for each SI parameters group in the network stabilize by iterations. Note also that ESS values tend to cluster around the same value, which is the main feature of the second regime.

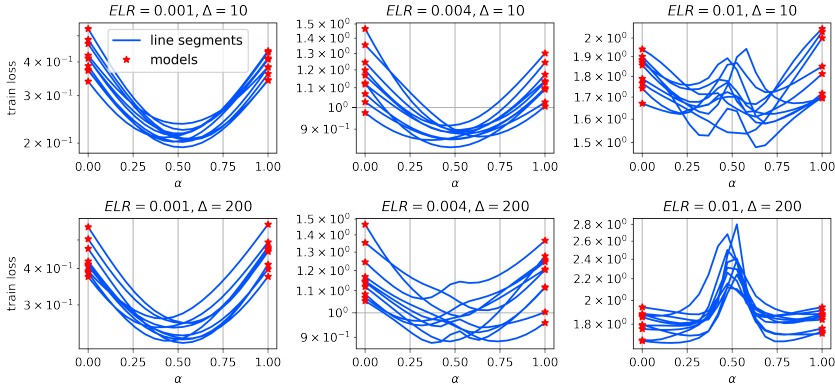

Figure 22: Second training regime. Linear connectivity of different checkpoints from the same training trajectory: small ELRs show locally convex loss, while large ELRs depict much more hilly landscape. $\Delta$ denotes the time difference (in epochs) between the connected checkpoints. ConvNet on CIFAR-10.

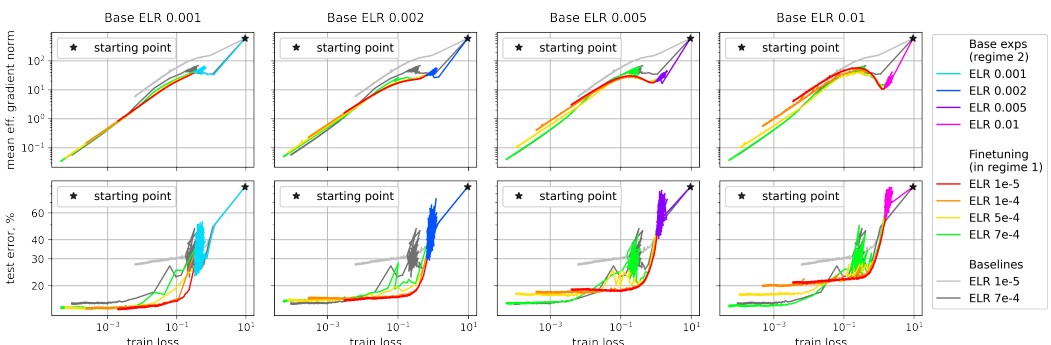

Figure 23: Second training regime. Fine-tuning with decreased ELR either keeps the same trajectory (for the lowest initial ELRs) or results in separate trajectories depending on the final ELR value (for the highest initial ELRs). ConvNet on CIFAR-10; fine-tuning for 2k epochs.

In Figure 21, we provide plots similar to Figure 5, left in the main text for other total ELR values. No equilibration of individual ESS values is observed neither in the first nor in the third regime, while in the second regime they accurately align (especially for the low ELRs of the second regime).

## K  Second regime: connecting checkpoints

In this appendix, we present the plots on linear connectivity of different checkpoints of the same trajectory in the second regime. From a given training trajectory obtained with a second regime ELR, we take several checkpoints and linearly connect them with the corresponding checkpoints obtained after $\Delta$ epochs. Figure 22 shows the loss behavior along these linear segments for several ELR and $\Delta$ values. We can notice that for small and moderate ELRs the training loss is almost convex and significantly lower in the middle of the segment than at its edges, which reinforces the "hopping along the walls" view on the second regime. For the highest ELRs of the second regime (especially when $\Delta$ is large), the loss is highly non-convex, indicating the more chaotic behavior of the optimization dynamics.

## L  Second regime: fine-tuning with first regime ELR

In this section, we provide an extensive comparison of fine-tuning results with lower ELRs starting from the second regime. As shown in Figure 23, top, the fine-tuning trajectories gradually tend to separate as the initial ELR increases (plots from left to right). In extreme cases, we have either

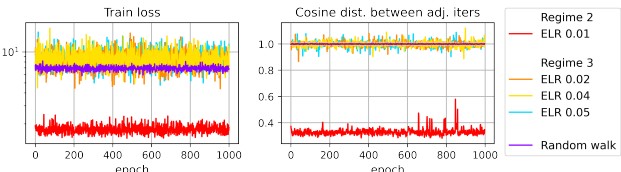

Figure 24: Random walk vs. the third regime dynamics. Cosine distance between weights on adjacent iterations (right) shows similarity between random walk and the third regime, while the training loss (left) shows difference. As a baseline, we show the largest ELR from the second regime. ConvNet on CIFAR-10.

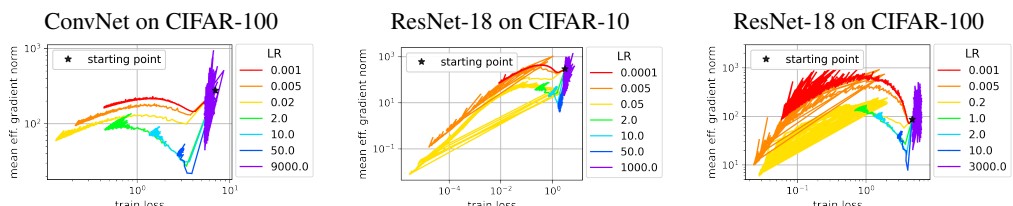

Figure 25: Three training regimes when training SI networks in the whole parameter space with weight decay.

convergence of all trajectories to the same wide optimum (lowest initial ELR 0.001), or a set of different trajectories covering the entire range of the first regime sharpness profiles (highest initial ELR 0.01). Therefore, we conclude that regions obtained with high second regime ELRs contain a rich spectrum of optima of various sharpness.

Figure 23, bottom, reveals the discrepancy between sharpness and generalization. For instance, while the fine-tuning trajectories starting from the highest ELR of the second regime closely match the baseline trajectories of the first regime in terms of sharpness range, the range of test errors achieved is much narrower than that of the baselines.

## M    Third regime: comparison with random walk

In this section, we compare the third regime dynamics with random walk (RW). For random walk dynamics, we train our model in the third regime but use a Gaussian random vector instead of a real gradient and appropriately scale its length to replicate the respective step size of a regular training. We found that the dynamics of RW corresponding to different ELRs in the third regime turn out to be very similar, therefore we show only one line for RW in Figure 24. We can see from Figure 24, right that the weights on adjacent iterations appear to be uncorrelated both in the third regime and in the RW case. This behavior substantially differs from the second regime, where the weights have relatively large correlation. Figure 24, left, demonstrates the difference between RW and the third regime: RW performs as a lower bound for the training loss achieved in the third regime.

## N    Three regimes in standard training

In this appendix, we present additional results for other dataset-architecture pairs complementary to Figure 7 in the main text.

When training SI networks in the whole parameter space using weight decay, we also observe the three regimes (Figure 25 supplements Figure 7, left). We note that ResNet-18 models demonstrate a periodic behavior in accordance with the results of Lobacheva et al. [27].

In the experiments with conventional training only first two regimes are present (Figure 26 supplements Figure 7, right). The regimes appear more distinguishable for ResNet-18 than for ConvNet (e.g., Figure 26, top right vs. bottom left), probably due to the fact that ResNet-18 is a significantly larger model. In the experiments with fixed LR, the best test accuracy is achieved at the largest

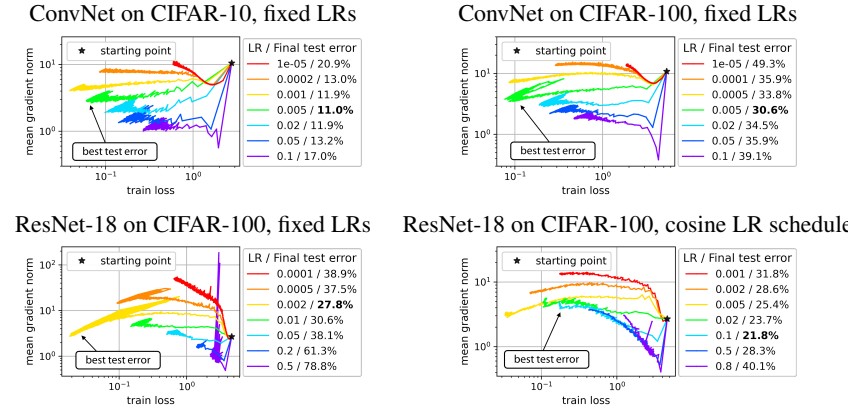

Figure 26: Training regimes in conventional training with fixed LRs and cosine LR schedules.

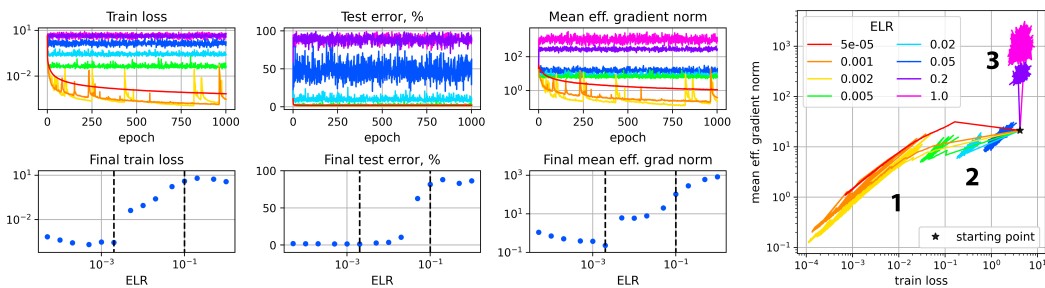

Figure 27: Training of fully-scale-invariant MLP on MNIST: all three training regimes are present.

LR of the first regime. We also conduct the experiment with cosine LR schedule for ResNet-18 on CIFAR-100 (see Figure 26, bottom right). As stated in the main text, the best performing models with cosine LR schedule correspond to the medium initial LR when both the first and second regimes are passed during the optimization.

## O    Additional experiments on other architectures and datasets

Most of our experiments include typical image classification tasks (CIFAR-10 and CIFAR-100) solved using convolutional architectures (ConvNet and ResNet-18). In this appendix, we present the results concerning the three training regimes obtained for other architectures and datasets: multilayer perceptron (MLP) on MNIST [7] and Transformer [34] on the AG NEWS dataset [39].

We train MLP on MNIST using the same setting as in the majority of our experiments, i.e., we make the network fully scale-invariant and optimize it on the sphere using projected SGD with a fixed ELR. We consider the following MLP architecture: a fully-connected neural network with two hidden layers of size 300 and 100, respectively. We use ReLU activation and add BN after both hidden linear layers. We made this network fully scale-invariant in accordance with Appendix C.

For the MLP on MNIST, which is shown in Figure 27, we can observe the same division on the three regimes: convergence to a minimum, chaotic equilibrium at some non-trivial loss level, and about random guess behavior. Some runs in the first regime exhibit a periodic behavior similar to the one observed with ResNet-18 in Appendix D due to similar reasons. We have also conducted experiments with MLP on the CIFAR-10 dataset but do not include them here, since the architecture appeared to be too simple to reach low loss values and adequately depict all three regimes (see Appendix F for a discussion of this issue).

In addition, we conducted an experiment in a domain other than computer vision in order to be more objective in our conclusions. We trained a Transformer neural network on the AG News classification dataset using standard SGD with weight decay of 1e-3, momentum of 0.9, and a constant learning

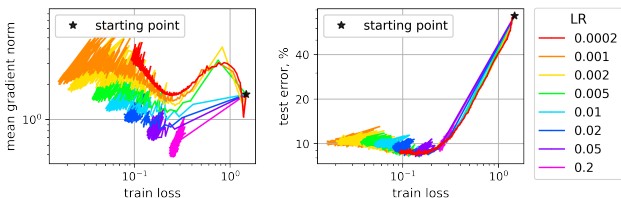

Figure 28: Conventional training of Transformer on the AG News. Results are similar to conventional training of ResNet-18 on CIFAR-10 in Section 6: the first two regimes can be clearly distinguished.

rate. For this dataset we used the encoder-only Transformer with $N = 1, d_{model} = 32, h = 1$. We slightly modified the original architecture to make it more scale-invariant (similar to the modifications we made in ResNet architecture): we add a LayerNorm [2] after each fully-connected layer in feed-forward sub-layers. Note that even after these changes the architecture is still not fully scale-invariant; making the Transformer architecture fully scale-invariant can be rather non-trivial [25], and we leave it for future work.

Sharpness/generalization vs. training loss diagrams for the Transformer on AG News are presented in Figure 28. The results are very reminiscent of the conventional training of ResNet-18 from Section 6. We can clearly observe the first two regimes on the diagram, while the third regime training quickly leads to NaNs. There is, however, a peculiar difference from the computer vision setting in a sense that the best achieved test error can be observed for LR values from both the first and the second regimes. We attribute this to overfitting of the model to the dataset.