# OpenReview forum: "Training Scale-Invariant Neural Networks on the Sphere Can Happen in Three Regimes"
_NeurIPS.cc/2022/Conference — NeurIPS 2022 Accept_

### Official Review · Reviewer_nwDp · 2022-07-10

**Rating:** 6
**Confidence:** 3
**Soundness:** 3 good
**Presentation:** 3 good
**Contribution:** 2 fair

**Summary:**

The paper investigates the directional optimization dynamics of training scale invariant neural networks. By fixing the effective learning rate (ELR), the paper seeks to decouple the influence of norm dynamics on the gradient from directional dynamics. The authors observe that training can behave according to three distinct regimes depending on the chosen ELR value. The various regimes are carefully analyzed both theoretically and empirically through the use of toy examples as well as more realistic settings. The authors utilize their framing to offer explanations to various conflicting observations in previous works, as well as share suggestions on how to use their insights for training more conventional neural networks.

**Questions:**

**Questions:**
- Section 3.1, starting line 102: what is the difference of the method described currently versus the statement ‘one learning rate for all spheres is overconstraining’? This difference still appears unclear after reading Appendix B
- Section 3.2, starting line 152: the analysis in this paragraph at some point switches from features of regime 1 to regime 2, yet it’s not quite clear exactly where this happens?
- Section 3.2, line 158/159: ‘is also non-vanishing (we give a more formal argument in the appendix)’ → how should one interpret the relevance of ‘non-vanishing’? Perhaps point to specific appendix?
- Section 3.2, starting line 171: for the toy example, it is clearly demonstrated how constant ELR are chosen to end up in one of the three regimes. Would the authors have a suggestion how this type of reasoning of picking ELR values could be extended to the more realistic settings beyond trial and error?
- Section 4 (related to the previous question): could the authors elaborate as to how the learning rates are chosen for the conducted experiments?
- Section 5.1: in the discussion surrounding picking different ELR values in the first regime, it is not entirely clear how the following concepts all move together: speed, loss, sharpness, robustness. Would it perhaps be possible to include a small table with this? E.g. rows: (speed, loss, sharpness, robustness), columns: 1st regime low ELR, 1st regime high ELR
- Section 6, starting line 336: could the authors elaborate why NaN values would eliminate the third regime in this setting compared to the previous settings?

**Suggestions:**
- [Throughout paper] sphere → hypersphere or n-sphere
- Section 3.1, equation 1: perhaps change R^P to S^(P-1) to indicate hyperspherical space?
- Section 3.2, line 122: ‘it can be shown’ → maybe point to relevant appendix or related work?

**Possible typos**:
- Section 3, line 87: ‘alike normalized neural networks’
- Section 3.2, line 125: ‘from what follows’ → ‘from which it follows’
- Section 3.2, line 156: ‘of individual’ → of the individual


**Limitations:**

The authors adequately addressed limitations of their work from a technical point of view. However, as pointed out in the '*weaknesses section*': (1) more practical advice around picking 'good' ELR values/ranges (or when this is possible/impossible), and (2) clearly stating hopes/expectations as to how/where this work could be improved upon would likely strengthen this work. Negative societal impact does not seem relevant.

**Strengths And Weaknesses:**

**Strengths**
- [**quality**] the main statements presented in the paper are generally well supported by clear, theoretical foundations accompanied by well explained proofs in the appendix.
- [**clarity**] the paper is clearly written and overall easy to follow. It also provides an extensive appendix containing additional discussion as well as further edge case experimental results to aid replication and completeness.
- [**significance**] the authors carefully analyzed a very specific setting that does not seem to have been explored in isolation before. However, because it touches on quite broad learning settings for neural network training - the (partial) use of scale-invariant modules - it could provide guidance and save time for other researchers meaning to conduct research in this topic, or practitioners analyzing observed learning behavior of their models.


**Weaknesses**
- [**clarity**] the provided plots are not always easy to interpret, as they are quite small with lines strongly overlapping. A possible suggestion to increase the legibility would be to enlarge the most important plots and move others to the appendix.
- [**significance**] while the paper does a great job at completing an isolated analysis seemingly absent from the relevant literature, it is not entirely clear to this reviewer how the provided results, i.e. the existence of three optimization regimes depending on the effective learning rate for (partially) scale-invariant NN, should be utilized in practice. Generally, the paper could be enhanced with more direct take-aways or suggestions as to how the results presented in this work should be used or build upon.

---

> ### Author Response · Authors · 2022-08-02
> **Response to Reviewer nwDp (1/2)**
>
> We are very thankful to the Reviewer for his/her comprehensive and attentive review on our work! We address the questions below one by one.
>
> > the provided plots are not always easy to interpret, as they are quite small with lines strongly overlapping. A possible suggestion to increase the legibility would be to enlarge the most important plots and move others to the appendix.
>
> That is a good suggestion, thanks! We agree that current plots may be too small. We will do our best to improve it in the following revision of the text.
>
> > while the paper does a great job at completing an isolated analysis seemingly absent from the relevant literature, it is not entirely clear to this reviewer how the provided results, i.e. the existence of three optimization regimes depending on the effective learning rate for (partially) scale-invariant NN, should be utilized in practice. Generally, the paper could be enhanced with more direct take-aways or suggestions as to how the results presented in this work should be used or build upon.
>
> Thank you for the remark, we will try to make the implications of our work more standout in the text.
> The main goal of our study was to investigate the peculiarities of the intrinsic domain of scale-invariant neural networks with careful control of the effective learning rate. In the paper, we developed theoretical and empirical groundings for distinguishing three regimes of training SI models on the sphere that shed more light on the intrinsic structure of their loss landscape (Section 5). Even though we mostly consider fully SI models, we demonstrated in Section 6 that our results are relevant for regular neural networks as well since normalization techniques make the majority of their parameters scale-invariant. In particular, we provided and explained from the perspective of our framework several practical takeaways that can be used to achieve the most optimal minima in terms of sharpness and generalization, e.g., taking the largest first regime LR to train a network with a fixed LR schedule and picking the medium one belonging to the second regime when training with a cosine schedule. We think that similar analysis of other LR schedules can also be an interesting future research direction, please see our general comment for further discussion.
>
> > Section 3.1, starting line 102: what is the difference of the method described currently versus the statement ‘one learning rate for all spheres is overconstraining’? This difference still appears unclear after reading Appendix B
>
> We are sorry for not making that clear enough, we will fix it in the text.
> As we state in Section 3.1, there can be many SI parameter groups in real networks since normalization techniques, like Batch Normalization, are usually applied multiple times in one network. One can fix the ELR for each group separately (project each group onto its own sphere) or consider them as one large SI group and project onto a single sphere, which is more feasible and closer to common practice (see Appendix B), and we do so in our work. The latter fixes the total ELR but allows the individual ones to adapt according to expression (5), which is crucial for our analysis. What we mean by ‘one learning rate for all spheres is overconstraining’ is that in the first way we could also mandatorily fix each individual ELR at a specific value, but that would completely hinder the adaptation of the individual ELRs and indeed overconstrain the optimization problem (if not make learning impossible).
>
> > Section 3.2, starting line 152: the analysis in this paragraph at some point switches from features of regime 1 to regime 2, yet it’s not quite clear exactly where this happens?
>
> We will write that more clearly. The ‘switch’ occurs right after equation (8), which sets the threshold for ELR: if it is below (8), then we are in the first regime, otherwise we are in the second one.
>
> > Section 3.2, line 158/159: ‘is also non-vanishing (we give a more formal argument in the appendix)’ → how should one interpret the relevance of ‘non-vanishing’? Perhaps point to specific appendix?
>
> We apologize for this inaccuracy, here we are talking about App. A.3, where we state this point more precisely. In fact, there we formally show why the effective step size values do not decay and the ELRs stabilize in equilibrium (9).

---

> ### Author Response · Authors · 2022-08-02
> **Response to Reviewer nwDp (2/2)**
>
> > Section 3.2, starting line 171: for the toy example, it is clearly demonstrated how constant ELR are chosen to end up in one of the three regimes. Would the authors have a suggestion how this type of reasoning of picking ELR values could be extended to the more realistic settings beyond trial and error?
>
> Unfortunately, such an exact analysis can only be carried out for a simplified setting. In real SI neural networks the boundary between the regimes can depend on the network architecture, dataset, etc. (cf. Fig. 1 in the main text and Fig. 11, top in the appendix). More general results on this are future work. However, based on the results of Section 5, it is possible to look at proxy indicators like ESS alignment, as in Fig. 5, left, to distinguish between the regimes and grope for the necessary ELR.
>
> > Section 4 (related to the previous question): could the authors elaborate as to how the learning rates are chosen for the conducted experiments?
>
> We started from the values of ELR, which are usually encountered during training of regular networks, and then expanded the range around them as much as possible. We also used a more fine grained ELR grid closer to the boundaries between the regimes.
>
> > Section 5.1: in the discussion surrounding picking different ELR values in the first regime, it is not entirely clear how the following concepts all move together: speed, loss, sharpness, robustness. Would it perhaps be possible to include a small table with this? E.g. rows: (speed, loss, sharpness, robustness), columns: 1st regime low ELR, 1st regime high ELR
>
> Thank you for the suggestion, we will try to include such a table in our text. Briefly speaking, in the first regime, higher ELR values result in faster training and better optima in terms of sharpness/generalization.
>
> > Section 6, starting line 336: could the authors elaborate why NaN values would eliminate the third regime in this setting compared to the previous settings?
>
> When experimenting with a fully scale-invariant network we fix the last layer weights (and other non-SI parameters), therefore the output of the network and thus its loss is always bounded. On the other hand, in the experiments with conventional training in Section 6, weights of the last layer can become large enough to lead to numerical overflows.
>
> > [Throughout paper] sphere → hypersphere or n-sphere
>
> Thanks for the suggestion but we use the accepted terminology in works on SI models (see, e.g., [1]).
>
> > Section 3.1, equation 1: perhaps change R^P to S^(P-1) to indicate hyperspherical space?
>
> Thank you, we will take that into account.
>
> > Section 3.2, line 122: ‘it can be shown’ → maybe point to relevant appendix or related work?
>
> Absolutely, we forgot to reference Appendix A.1, thanks a lot!
>
> > Possible typos
>
> Many thanks, we will fix it in the text.
>
> [1] Wan, R., Zhu, Z., Zhang, X., and Sun, J. (2021). Spherical motion dynamics: Learning dynamics of normalized neural network using sgd and weight decay. In Ranzato, M., Beygelzimer, A., Dauphin, Y., Liang, P., and Vaughan, J. W., editors, Advances in Neural Information Processing Systems, volume 34, pages 6380–6391. Curran Associates, Inc.

---

### Official Review · Reviewer_UmdY · 2022-07-10

**Rating:** 5
**Confidence:** 4
**Soundness:** 3 good
**Presentation:** 4 excellent
**Contribution:** 2 fair

**Summary:**

This paper studies the training behavior of scaling invariant neural networks using projected SGD with fixed effective learning rate on the sphere. The topic is meaningful because neural network models become (semi) scale invariant when normalization techniques are applied. Theoretically, the authors point out several relations satisfied by the effective learning rates and effective step sizes of individual parameter groups and the whole parameter vector. With the relations, the dynamics of individual parameter groups' effective learning rates are derive, showing a "negative feedback" principal, which is later used to explain the "chaotic equilibrium" regime of training. Also theoretically, a simple scale invariant function is analyzed to study the different optimization behaviors shown by different effective learning rate.

Empirically, experiments are conducted on both scale invariant neural networks and regular neural networks. Three regimes are identified for the behavior of the training loss: convergence, chaotic equilibrium, and divergence. The three regimes happen in order when the effective learning rate is increased. Extensive experiments are conducted to observe and compare the test error and sharpness of solutions for the three regimes. Discussions are made trying to explain the three regimes using the theoretical results.

**Questions:**

1. It there any difference in the training behavior of scale invariant networks and regular networks? If such differences exist, are they able to be explained by the theoretical insights obtained in this paper?

2. For the toy function (7), it seems there no divergence regime by proposition 1. Theoretically understanding, what happens when the behavior moves from regime 2 to regime 3?

3. The first line of equation (1) is confusing.

**Limitations:**

Limitations are discussed in the last section.

**Strengths And Weaknesses:**

Originality: The theoretical results obtained in Section 3 is new. Also, the extensive experiments on scaling invariant neural networks with fixed effective learning rate are new. Though, the reviewer does not think the observations dramatically change our understanding on neural networks' training process. It is a common belief that, even for non-scale-invariant networks, the loss experiences smooth descending, oscillation, and divergence when the learning rate is increased. The experiments and resulting observations on the sharpness of solutions, and jump between basins, also exist in previous works. The results for scale invariant networks are not different. Also, the theoretical results are not sufficient and convincing enough to explain the empirical behaviors.

Quality and Clarity: The paper is well written and easy to follow. The descriptions and discussion on numerical experiments are clear.

Significance: The work has fair significance on our understanding to neural networks with normalizations. It shows some widely known phenomena in training also hold for scale invariant models trained with fixed effective learning rate.

---

> ### Author Response · Authors · 2022-08-02
> **Response to Reviewer UmdY (1/2)**
>
> We thank the Reviewer for his/her detailed and thoughtful review! We now address the main concern about the originality of our work as well as the questions of the Reviewer.
>
> ### On originality of our work
> We fully agree with the Reviewer that some of the results presented in our paper have already been known to some extent in the community, e.g., that training with larger learning rates typically results in better optima, which however still does not seem as a limitation of our work but rather gives an additional confirmation of the conventional wisdom. Nevertheless, we believe that the major contribution of our work — careful study of the loss landscape of SI neural networks on their intrinsic domain (Section 5) — is novel and important with an impact on regular neural networks training (Section 6) since normalization techniques are ubiquitous in practice. As the Reviewer has rightly remarked, we are the first to study the optimization behavior of scale-invariant models with a fixed effective learning rate. By doing so, we manage to eliminate the undesirable effects of the changing optimization speed that precluded adequate studying of the properties of their intrinsic domain (unit sphere) and led to controversial reports in literature (please see Sections 1 and 2 for a more detailed discussion).
>
> To be more concrete, we would like to give a direct comment to some quotations from the review.
>
> > It is a common belief that, even for non-scale-invariant networks, the loss experiences smooth descending, oscillation, and divergence when the learning rate is increased.
>
> As far as we know, previous analysis of training regular neural networks has mainly reported only *two* basic regimes depending on the value of the learning rate: either convergence or complete divergence [1]. The convergence can indeed be non-monotonic [2], even unstable [3] but it is far from the pronounced chaotic equilibrium regime that we discovered. Even though the second regime can sometimes be observed in conventional training, as we show in Section 6, it is still difficult to distinguish it from the first regime, which makes it deceptive to believe that they are the same. In fact, the first two regimes depict completely different properties of the loss landscape (Section 5) and consequences for the training (Section 6) due to the presence of a phase transition while traversing the high-sharpness zone of the landscape. It is only in our setup, when all distractions such as the variable ELR are removed, that the boundary between the regimes becomes so clear.
>
> > The experiments and resulting observations on the sharpness of solutions, and jump between basins, also exist in previous works.
>
> To the best of our knowledge, previous works only sparingly touched upon the properties of the final solutions and the training trajectory when finetuning with smaller and (especially) larger learning rates. In this regard, our work is the first one to comprehensively conduct such experiments while taking into account the training regime and intrinsic domain of the models.
>
> > Also, the theoretical results are not sufficient and convincing enough to explain the empirical behaviors.
>
> We admit that current theory is not capable of explaining all the empirical results obtained when training scale-invariant neural networks with projected SGD due to the difficulty of the problem. However, we believe that our theoretical results provide some guidance into the observed behavior by, firstly, analyzing a model example and, secondly, predicting certain characteristics of the regimes based on the general principles (like ESS equilibration in the second regime, see Figure 5, left). Developing a more general and sophisticated theory is a promising future research direction.

---

> ### Author Response · Authors · 2022-08-02
> **Response to Reviewer UmdY (2/2)**
>
> We now answer the specific questions asked by the Reviewer.
>
> > It there any difference in the training behavior of scale invariant networks and regular networks? If such differences exist, are they able to be explained by the theoretical insights obtained in this paper?
>
> Since using normalization techniques has become a common practice in deep learning, almost all modern neural networks contain scale-invariant parameters. The training process of neural networks without normalization layers is much more unstable; in particular, we could not observe a persistent second regime for such networks because training with relatively high LRs quickly fails with NaN weights. If we compare partially and fully SI networks, the latter can be more stable to train and more convenient to study SI-specific properties of neural networks training (Section 2), while still performing on par with the former; in particular, due to fixation of the last layer, SI networks almost never fail with NaNs during training. However, SI models can exhibit some peculiar behavior during training as well; please see Lobacheva et al. [4] for more detail and further comparison of training SI and regular NNs. Still, our work does not intend to point at and explain the differences in training regular and SI networks but instead tries to develop insights into the properties of the intrinsic domain of SI models and transfer them to the conventional networks training since, again, many (if not most) of modern networks parameters are normalized and hence scale-invariant.
>
> > For the toy function (7), it seems there no divergence regime by proposition 1. Theoretically understanding, what happens when the behavior moves from regime 2 to regime 3?
>
> In our work, we draw the line between the 2nd and the 3rd regimes by comparing them with random guess quality (see Appendix L): the third regime is much more reminiscent of the random walk behavior than the second one (chaos >> equilibrium). Using the same criterion, we may notice that in Proposition 1 in the limit of $\tilde{\eta} \to \infty$ the value of the function $f_{\alpha}(x, y)$ around which optimization “stabilizes” is $\frac{\alpha}{2}$ corresponding to the expected value of the function given that input points are randomly sampled on the sphere, which totally accords with our intuition. Figure 2 demonstrates this transition from chaotic equilibrium to total chaos.
>
> > The first line of equation (1) is confusing.
>
> This is a shortcoming of our notation. We will fix it in the following revision of our paper, thank you for the remark.
>
> [1] Gilmer, J., Ghorbani, B., Garg, A., Kudugunta, S., Neyshabur, B., Cardoze, D., Dahl, G. E., Nado, Z., and Firat, O. (2022). A loss curvature perspective on training instabilities of deep learning models. In International Conference on Learning Representations.
> [2] Cohen, J., Kaur, S., Li, Y., Kolter, J. Z., and Talwalkar, A. (2021). Gradient descent on neural networks typically occurs at the edge of stability. In International Conference on Learning Representations.
> [3] Lewkowycz, A., Bahri, Y., Dyer, E., Sohl-Dickstein, J., and Gur-Ari, G. (2020). The large learning rate phase of deep learning: the catapult mechanism.
> [4] Lobacheva, E., Kodryan, M., Chirkova, N., Malinin, A., and Vetrov, D. P. (2021). On the periodic behavior of neural network training with batch normalization and weight decay. In Ranzato, M., Beygelzimer, A., Dauphin, Y., Liang, P., and Vaughan, J. W., editors, Advances in Neural Information Processing Systems, volume 34, pages 21545–21556. Curran Associates, Inc.

---

> > ### Comment · Reviewer_UmdY · 2022-08-08
> > **Tend to accept now**
> >
> > Thanks to the detailed response from the authors. Now I would like to increase my score because the following point that I missed before:
> >
> > In practice, most networks contain normalization layers. Therefore, many the behaviors of "regular networks" I referred to in the original review are actually behaviors of the normalized networks. This point does not stand as a strong point to criticize this work.

---

> > > ### Author Response · Authors · 2022-08-08
> > > **Thank you!**
> > >
> > > We would like to express our gratitude to the Reviewer for his/her decision to raise the score! We are sure that taking into account the comments received will significantly improve the positioning and clarity of our work.

---

### Official Review · Reviewer_AvYP · 2022-07-11

**Rating:** 7
**Confidence:** 3
**Soundness:** 4 excellent
**Presentation:** 3 good
**Contribution:** 3 good

**Summary:**

This paper studies properties of training scale invariant nets using fixed learning rates. In this setting, it further characterized 3 distinct regimes where qualitative behavior NN training and generalization is different. These properties hold both in an illustrative toy problem and practical neural nets.

**Questions:**

I like the paper overall, I'm curious if the authors can suggest future avenues of direction if and how these insights can be used to improve either training or generalization performance of practical (maybe scale invariant) neural nets.

**Limitations:**

Yes

**Strengths And Weaknesses:**

Strengths
- Solid empirical results and conclusions follow directly from the evidence
- The paper also does a good job of setting up the background and situating itself with related work.
- Toy problem is not novel but is indeed illustrative

Weaknesses
- Figures are too small, I would encourage the authors to make them bigger for improved reading experience

---

> ### Author Response · Authors · 2022-08-02
> **Response to Reviewer AvYP**
>
> First of all, we want to thank the Reviewer for his/her feedback and overall positive assessment of our work! We are glad that the Reviewer finds our results solid and illustrative. We respond to the comments as follows.
>
> > Figures are too small, I would encourage the authors to make them bigger for improved reading experience
>
> It is indeed a drawback of the current version of our text and we will do our best to improve the appearance of our figures, thank you.
>
> > … I'm curious if the authors can suggest future avenues of direction if and how these insights can be used to improve either training or generalization performance of practical (maybe scale invariant) neural nets.
>
> In the paper, we discuss several practical implications of our analysis both for scale-invariant (Section 5) and regular neural networks (Section 6). For instance, we find that the best choice of the initial learning rate in terms of the final test performance is the largest first regime one for the fixed LR schedule and the medium second regime one for the cosine LR schedule. We also think that our insights may help to analyze and construct better LR schedules in the future (please see our general comment for more discussion).

---

### Official Review · Reviewer_FkbE · 2022-07-13

**Rating:** 8
**Confidence:** 3
**Soundness:** 4 excellent
**Presentation:** 3 good
**Contribution:** 4 excellent

**Summary:**

The authors perform a rigorous study of training regimes for neural networks with normalized weights, which is a common practice. There is a mathematical analysis, using an example problem, and empirical study on standard image classification benchmarks. The paper culminates with a comparison of training regimes on a Resnet18 on cifar10 trained with and without scale invariance.

**Questions:**

# Section 6 questions

The experiment setup in section 6 needs some clarification. What exactly is teh difference between the three cases in Figure 7?
What exactly does it mean to do scale invariance “in the whole parameter space”? How is this different from the models in the previous sections?
To clarify, is spherical projection applied during training for these models? If so, for which?
Why did you vary LR instead of ELR?
How are batchnorms applied in all three cases? Does the ResNet 18 contain batch normalization layers in the conventional training runs?
Figure 7: Why is the test error missing on the rightmost SI figure? Did these networks perform worse?
Which of the methods in Figure 7 is “the best” in terms of accuracy, training performance, robustness, etc.?
Line 340: How does data augmentation make the problem “too hard”? Shouldn’t it make the problem easier?

# Clarification points

The first line of the paper calls out batch normalization and layer normalization but these methods normalize hidden states, not the weights themselves. Spherical projection normalization is different. To which normalization techniques exactly does the analysis apply? The first paragraph would be better to clarify this.

The text needs better descriptions of architectures. What architecture is “ConvNet” exactly? What are the activations, layer width, etc.

# Discussion questions:

Given these results, what observation can I make in practice to inform decisions on model training? Do the results suggest a way of determining scale invariant weights will improve my model?

Does your work show any advantages to using scale invariant weights? Are there any roadblocks that your analysis removes from using scale invariant weights in practice?

# Minor

Figure 7: I first read “Conv. training” and “convolutional” instead of “conventional”; it’s a confusing overload of an abbreviation.

Extra space in line 3 of Figure 7 caption , .


**Limitations:**

The text addresses the limitations of the work. As an empirical understanding paper, there are no major limitations I see, and it is a contribution as-is.

**Strengths And Weaknesses:**

The analysis is thorough, and paying attention to normalization methods makes this paper a solid contribution. Practitioners are more familiar with “normal” training techniques, but in the real world, spherical projections are common, but not as well understood. The authors make a good case that the training behavior changes significantly, and the presentation which will help readers build intuition to interpret these in practice. Section 6 which compare-and-contrasts is exceptional at achieving that end.

One weakness is that the paper only focused on CIFAR-10/0 with convolutional nets. These problems are over-optimized that batch norm alone works. It would be interesting to see if the analysis applied to other domains and architectures, such as MLPs or transformers, where spherical projection is more common. That would maximize the paper’s impact, but it is substantial as-is.

---

> ### Author Response · Authors · 2022-08-02
> **Response to Reviewer FkbE (1/2)**
>
> We are very grateful to the Reviewer for his/her positive, detailed and constructive feedback on our work! We address the questions below one by one.
>
> > It would be interesting to see if the analysis applied to other domains and architectures, such as MLPs or transformers, where spherical projection is more common.
>
> We totally agree with the Reviewer on this point. We conducted additional experiments with a fully-connected neural network and revealed the same division on the three regimes depending on the ELR value. We used a version of LeNet-300-100 with BN after each layer on the MNIST dataset. To make the network fully scale-invariant we fixed the BN parameters and the last layer weights in the same way as we did for ConvNet in the paper. Construction of a fully scale-invariant transformer is not that straightforward due to the order of normalization layers and residual connections in the standard architecture, therefore we currently leave it for future work.
>
> > What exactly is the difference between the three cases in Figure 7?
>
> As we write in Section 6, the leftmost subfigure of Figure 7 represents training a fully scale-invariant neural network (with all non-scale-invariant parameters fixed) using standard SGD with weight decay without making a projection onto the sphere, the middle one is for training a standard neural network with both scale-invariant and non-scale-invariant parameters using a standard training procedure with a fixed learning rate, and the rightmost one depicts training a standard neural network with cosine LR schedule.
>
> > What exactly does it mean to do scale invariance “in the whole parameter space”? How is this different from the models in the previous sections? To clarify, is spherical projection applied during training for these models? If so, for which? Why did you vary LR instead of ELR?
>
> The models in sections prior to Section 6 a) are fully scale-invariant, i.e., all of their trainable parameters precede the normalization layers, which makes these models invariant to rescaling their weights by a positive constant, and b) were trained using a projected SGD method, i.e., after each gradient step the network’s parameters were projected onto the sphere, which allowed for a more subtle control of their effective learning rate. In Section 6, we showed that our results about the existence of three regimes of training SI models on the sphere can be extended to more practical scenarios, in particular, when training fully scale-invariant neural networks using standard SGD with weight decay in the whole parameters space, i.e., *without projecting the weights onto the sphere after each gradient step* (leftmost plot in Figure 7). Since we apply a standard SGD in this case, we do not control the ELR but instead sweep through LR values, which is much more common in practice.
>
> > How are batchnorms applied in all three cases?
>
> The Batch Normalization is applied in the same way in all the experiments, except we fix the learnable affine parameters in BN layers in case of fully scale-invariant networks since these parameters are non-scale-invariant.
>
> > Does the ResNet 18 contain batch normalization layers in the conventional training runs?
>
> Yes, more details on the concrete implementation can be found in Appendix D. All networks in all experiments in the paper contain BN layers.
>
> > Figure 7: Why is the test error missing on the rightmost SI figure? Did these networks perform worse?
>
> We suppose the Reviewer meant the leftmost plot in Figure 7 (SI networks in the whole space). Networks in this experiment indeed perform worse because we do not use data augmentation during training. The best test error of 15.15% is achieved with LR 2.0 (yellow line). The goal of this particular experiment was not to achieve the best performance, but to show that training of SI networks in the whole parameter space clearly happens in the same three regimes as we observed when trained on the sphere.
>
> > Which of the methods in Figure 7 is “the best'' in terms of accuracy, training performance, robustness, etc.?
>
> Conventional training with cosine schedule is the best one, expectedly (rightmost plot in Figure 7). Moreover, as we discussed in Section 6, the best results are obtained with cosine schedule starting from the medium LR values since in this case training can localize a region with the flattest optima while training with the second regime LRs and then has just enough time to converge to the minimum with the first regime LR at the end. In the leftmost plot, training does not achieve optimal results due to the lack of data augmentation, which is crucial for generalization. In the middle plot, training also cannot achieve the best results because it lacks the flexibility given by a proper LR schedule.

---

> ### Author Response · Authors · 2022-08-02
> **Response to Reviewer FkbE (2/2)**
>
> > Line 340: How does data augmentation make the problem “too hard”? Shouldn’t it make the problem easier?
>
> Data augmentation makes the original optimization problem of learning the given dataset much more difficult since it increases the complexity of datasets. In these terms achieving low values of the training loss becomes harder and a larger neural network is often required. Please see Appendix G for more details on the subject.
>
> > The first line of the paper calls out batch normalization and layer normalization but these methods normalize hidden states, not the weights themselves. Spherical projection normalization is different. To which normalization techniques exactly does the analysis apply? The first paragraph would be better to clarify this.
>
> Our analysis is applicable to any normalization technique that makes (a part of) neural networks weights scale-invariant, i.e., the model’s output does not change after multiplying these parameters by a positive constant. These techniques include but are not limited to Batch Normalization, Layer Normalization, Weight Normalization, etc. Since SI models are invariant to rescaling of their parameters, we can consider them as intrinsically defined on the sphere. Thus, our main point was to thoroughly investigate the properties of training SI neural networks on the sphere directly and then compare our results with the existing views on general neural networks training.
>
> > The text needs better descriptions of architectures. What architecture is “ConvNet” exactly? What are the activations, layer width, etc.
>
> As we stated in Section 4 and Appendix D, we use concrete realizations of the mentioned architectures from the following repository: https://github.com/tipt0p/periodic_behavior_bn_wd. ConvNet contains a first convolutional layer with k filters followed by BN and ReLU, then three blocks of convolution + BN + ReLU + maxpool with 2k/4k/8k filters in convolutional layer, and then maxpool + linear layer. In most of the experiments, we use ConvNet with a width factor k equal to 32. We will provide more detail in the appendix in the next revision of the text.
>
> > Given these results, what observation can I make in practice to inform decisions on model training? Do the results suggest a way of determining scale invariant weights will improve my model?
>
> We give some practical suggestions from our analysis in Section 6. For instance, it is most beneficial to select the largest learning rate value corresponding to the first regime when training with a constant LR and choose medium values from the second regime when using standard cosine annealing. We also think that the training properties we discovered in the paper may help to analyze and construct better LR schedules in the future.
>
> > Does your work show any advantages to using scale invariant weights? Are there any roadblocks that your analysis removes from using scale invariant weights in practice?
>
> Scale-invariant weights are always present in neural networks that use normalization (e.g., Batch Normalization), which has become a common practice nowadays. Speaking of a projected SGD method that we use in our analysis, it may not significantly improve (or worsen) the final test performance of the SI model but allows us to more straightforwardly study the intrinsic domain of such models by directly controlling the effective learning rate value.
>
> > Figure 7: I first read “Conv. training” and “convolutional” instead of “conventional”; it’s a confusing overload of an abbreviation.
>
> > Extra space in line 3 of Figure 7 caption
>
> Thanks, we will fix that in the text.

---

### Author Response · Authors · 2022-08-02
**General Comment by the Authors**

Primarily, we want to thank all the Reviewers for their invaluable contribution to improving our paper! We were happy to read the positive comments as well as the constructive critique of our work. We strongly believe that our paper can significantly benefit from taking the answers to the questions of the Reviewers into account in the next revision of the text, and we will do our best to do so.
Below we would like to highlight the points that were mentioned in most of our responses, so we consider them the most relevant and specifically indicate them in this general comment.

### Main implications of our work
Since we have encountered questions about our contributions and implications (especially from the practical point of view), we decided to summarize them here. We will also improve our formulations in the next revision of the text.
Our study was primarily aimed to investigate the intrinsic domain (unit sphere) of scale-invariant neural networks using a manual control of the effective learning rate, which had not been done previously. We discovered three regimes of training SI models with a fixed ELR using projected SGD. Each of these regimes possesses its own specific properties and reveals the peculiarities of the optimization dynamics and intrinsic structure of the loss landscape. Our findings both reconfirmed some of the previous beliefs, like that training with larger (E)LR generally ends up in better optima, or that high-sharpness zones are present in the optimization trajectory, and introduced novel insights, like that different (E)LRs of the first regime even with the same initialization and batch ordering actually lead to distinguishable minima with different sharpness/generalization profiles depending on a concrete (E)LR value, or that the high-sharpness zones are responsible for the boundary between the 1st and the 2nd regimes and have close ties with the Double Descent phenomenon.
Investigating SI models is interesting by itself, but it also has a direct implication for practical deep learning since normalization methods in regular neural networks are typical and make most of the parameters scale-invariant. In Section 6, we specifically demonstrated how the results we obtained for SI models are reflected in conventional training. Thus, we have presented and interpreted in our framework several practical takeaways that can be used to achieve the most optimal minima in terms of sharpness and generalization, like taking the largest LR of the first regime to train a network with a fixed LR schedule and choosing the medium one from the second regime when learning with cosine LR annealing.

### Possible future directions
As several Reviewers have questioned us about the future directions, here we would like to briefly discuss some of the potential prospects for the ensuing research. First, it would be interesting to develop more solid and general theoretical groundings for the observed phenomena. For instance, a closer look at the negative feedback principle in eq. (5) could potentially shed more light on the ESS/ELR dynamics in the first and second regimes compared to the current decay vs. alignment behavior and maybe better explain occasional transitions from the second to the first regime for certain ELRs in Section 5.3. Second, we plan to extend our analysis to other tasks, architectures, and especially optimizers like Adam, which can be challenging due to their indirect impact on the effective direction and effective learning rate in SI models [1]. Third, it is of course very tempting to investigate other potential practical implications of our results for conventional NN training. We believe that the analysis of various LR schedules from the perspective of the discovered training regimes is intriguing and might help in designing more efficient and more explainable LR schedules. For example, from our experiments we observe that decreasing (E)LR in the first regime during training does not influence the final solution but only slows down the convergence; at the same time, for the optimal final performance, optimization should run through the low second regime (E)LRs, which allows to achieve basins containing solutions with the best sharpness/generalization properties.

[1] Roburin, S., de Mont-Marin, Y., Bursuc, A., Marlet, R., Pérez, P., and Aubry, M. (2020). A spherical analysis of adam with batch normalization. arXiv preprint arXiv:2006.13382.

---

### Meta-Review · Area_Chair_xQD2 · 2022-09-02

**Recommendation:** Accept
**Confidence:** Less certain

**Metareview:**

All reviewers find the papers analysis on optimization regimes for scale invariant networks interesting and observations novel. The results are clearly presented and well supported by the (limited) analysis. However the reviewers also highlight several drawbacks
1) the paper mainly presents analysis for a scalar function
2) experiments limited to Cifar datasets with ConvNets.

While the results for presented settings are convincing, I think it is hard to judge the universality/importance of the phenomenon from analysis of scalar functions and experiments only on Cifar. Overall I think the paper is on borderline and suggest acceptance as the phenomenon is well presented and can lead to further works building on this. I encourage authors to include more experiments on different datasets and model architectures in the final version.

**Award:**

No

---

### Decision · Program_Chairs · 2022-09-14

Accept